# Pervasive cold-ice within a temperate glacier - implications for glacier thermal regime, sediment transport and foreland geomorphology

Benedict T. I. Reinardy[1], Adam Booth[2], Anna Hughes[3], Clare M. Boston[4], Henning Åkesson[5], Jostein Bakke[3], Atle Nesje[3], Rianne H. Giesen[6*], Danni M. Pearce[7]

[1]Department of Physical Geography and Bolin Centre for Climate Research, Stockholm University, Stockholm, 106 91, Sweden
[2]School of Earth and Environment, University of Leeds, Leeds, LS2 9JT, UK
[3]Department of Earth Sciences, University of Bergen and Bjerknes Centre for Climate Research, Bergen, 5020, Norway
[4]Department of Geography, University of Portsmouth, Portsmouth, PO1 3HE, UK
[5]Department of Geological Sciences and Bolin Centre for Climate Research, Stockholm University, Stockholm, 106 91, Sweden
[6]Institute for Marine and Atmospheric research Utrecht, Utrecht University, Utrecht, 3508 TA, The Netherlands
[7]Department of Biological and Environmental Sciences, University of Hertfordshire, Hatfield, AL10 9AB, UK

*Correspondence to*: Benedict T. I. Reinardy (benedict.reinardy@natgeo.su.se)

[*]*Now at:* HydroLogic Research, Delft, The Netherlands

**Abstract.** This study suggests that cold-ice processes may be more widespread than previously assumed, even within temperate glacial systems. We present the first systematic mapping of cold-ice at the snout of the temperate glacier Midtdalsbreen an outlet of the Hardangerjøkulen icefield (Norway) from 43 line-kilometres of ground penetrating radar data. Results show a 40 m-wide cold-ice zone within the majority of the glacier snout, where ice thickness is <10 m. We interpret ice to be cold-based across this zone, consistent with basal freeze-on processes involved in the deposition of moraines. We also find at least two zones of cold-ice up to 15 m thick within the ablation area, occasionally extending to the glacier bed. There are two further zones of cold-ice up to 30 m thick in the accumulation area, also extending to the glacier bed. Cold-ice zones in the ablation area tend to correspond to areas of the glacier that are covered by late-lying seasonal snow patches that reoccur over multiple years. Subglacial topography and the location of the freezing isotherm within the glacier and underlying subglacial strata likely influence transport and supply of supraglacial debris and formation of controlled moraines. The wider implication of this study is the possibility that with continued climate warming, temperate environments with primarily temperate glaciers could become polythermal in forthcoming decades with i) persisting thinning and ii) retreat to higher altitudes where subglacial permafrost could be and/or become more widespread. Adversely, the number and size of late-lying snow patches in ablation areas may decrease and thereby reduce the extent of cold-ice, reinforcing the postulated change of thermal regime.

# 1 Introduction

Glaciers are generally divided into three types of thermal regime depending on whether the ice is at or below pressure-melting point, or a combination of the two states, i.e. temperate, polar and polythermal. The thermal regime at the ice-bed interface is referred to as the basal thermal regime and areas of a glacier bed can be described as either warm-based or cold-based. Thermal and basal thermal regime influences glacier surface mass balance and dynamics, and therefore affects processes of sediment delivery to the ice margin, as well as landform development. Increasingly, glacier thermal (and basal) regimes are more complex than previously assumed, and the actual thermal conditions of any individual glacier can be highly spatially variable (e.g. Rippin et al., 2011; Waller et al., 2012). A wide range of thermal structures are observed in polythermal glaciers (e.g. Blatter and Hutter 1991; Pettersson, 2004) varying from those containing a perennial cold surface layer overlying a temperate core (Pettersson et al., 2007) to high Arctic ice caps and icefields with cold-ice margins and ice frozen to the bed (basal freeze-on) (Dowdeswell et al., 1999). Past Pleistocene ice sheets are also known to have had frozen-bed patches (Kleman and Hättestrand, 1999), and in some situations, glacier thermal regime has been interpreted as a remnant reflecting past rather than present climatic regimes (Blatter and Hutter; 1991; Rippin et al., 2011). Cold-ice (ice that does not contain any liquid water below pressure melting point) may in fact also be present in isolated pockets within glaciers that are typically classified as temperate. Several geomorphological and sedimentological studies have suggested that some temperate glaciers may have a cold-ice margin. Basal freeze-on occurs particularly during winter when liquid water content at the bed is diminished and energy production through friction and internal deformation is reduced. Sediments are entrained at the base of the glacier snout and then melt out during the ablation season (Sharp, 1984; Krüger, 1993; 1994; 1995; 1996; Matthews et al., 1995; Krüger et al., 2002; 2010; Bradwell, 2004; Evans and Hiemstra, 2005; Winkler and Matthews, 2010; Reinardy et al., 2013; Hiemstra et al., 2015). This growing body of evidence suggests that cold-ice processes within primarily temperate-glacier systems may influence marginal depositional processes and landforms. In addition to this influence, to date, the role of cold-ice in glacier mass balance and flow regime of temperate glacier systems has been viewed as a minor component. However, this view is supported by little evidence from direct observations of cold-ice and its distribution within otherwise temperate glaciers.

Here, we determine, through systematic mapping, the thermal regime of Midtdalsbreen, a Norwegian glacier previously described as temperate (Andersen and Sollid, 1971) but known to have cold-based ice at the glacier margin (Hagen, 1978; Liestøl and Sollid, 1980; Etzelmüller and Hagen, 2005). Midtdalsbreen is an outlet glacier of the Hardangerjøkulen icefield in southern Norway (Fig. 1) that has been retreating from its Little Ice Age (LIA, cf. 1850) maximum position until the present day, except for a small regional readvance of glaciers across Scandinavia, including Midtdalsbreen, recorded during the 1990s (Nesje and Mathews, 2012). Detailed mapping of minor moraines (typically less than 2 m high) deposited during the 1960s and 1970s indicated that many were ice-cored (Østrem, 1964; Andersen and Sollid, 1971). A second survey of the glacier foreland (Reinardy et al., 2013) indicated that minor moraines deposited in the early 2000s were no longer ice-cored. Instead, sedimentological investigation of the moraines suggested basal freeze-on of subglacial sediments and incremental sediment

slab stacking mimicking the underside of the glacier snout. Reinardy et al. (2013) therefore suggested that at least part of the snout of Midtdalsbreen was cold-based. This in turn indicated that the snout of Midtdalsbreen was thin enough to allow winter cold to penetrate to the bed. During the brief (2-4 months) melt season, the bed directly below the glacier snout would remain below pressure melting point whereas sediments at the glacier margin would melt out to form "stacked" minor moraines. If the glacier was cold-based, even in part, during the last two decades of relative warmth when these moraines were formed, then similar cold-based conditions are likely to have persisted at least through the last 40-50 years when the climate was comparatively colder and even more conducive to basal freeze-on. This is particularly significant because the decade 2000-2010 is likely to have been the warmest since the LIA, reflected in increasing glacier retreat rates (Andresen, 2011; Andreassen et al., 2016), yet the sedimentological evidence within the moraines implies that the snout of Midtdalsbreen would have remained frozen throughout retreat. The potential role of cold-ice processes at the margins of palaeo-ice sheets (e.g. Boulton, 1972; Budd et al., 1976; Moran et al., 1980; Mooers, 1990) and frozen-bed patches below palaeo-ice sheets (e.g. Kleman and Borgström, 1994; Astakhov et al., 1996; Kleman and Häterstrand, 1999) has long been recognised. In addition, basal freeze-on processes have also been used to interpret the stagnation of ice streams (Christoffersen and Tulaczyk, 2003), glacier surge dynamics (Murray et al., 2000), bed strength (Meyer et al., 2018) and tunnel valley infill (Kristensen et al., 2008). However, the role of cold-ice processes within temperate glacier systems is relatively unknown.

We first investigate whether interpretations of cold-based ice and production of debris-rich basal ice from sedimentological evidence (Reinardy et al., 2013) are supported by observations of thermal regime using ground-penetrating radar (GPR) over the glacier snout. Second, we discuss the influence that the glacial and basal thermal regime may have on the production and transport of debris at the glacier snout and margin and the resulting geomorphic imprint preserved in the glacier foreland. Determining the detailed thermal regime of Midtdalsbreen is associated with three wider hypotheses: i) Cold-based ice and freeze-on processes can occur locally beneath glaciers that are otherwise classified as 'temperate', and can be evidenced from moraine sedimentology and geomorphology; ii) the conventional association of cold-based polar glaciers with a lack of glacial landforms should be reassessed since 'temperate' glaciers with cold-based margins may leave an equally subtle geomorphic imprint; iii) cold-based processes, including basal freeze-on, may have a significant role for the dynamics of temperate glaciers that are thinning and retreating to higher altitudes (even during, e.g. the exceptional climatic warming observed since the year 2000). Implications from these hypothesis suggest that cold-based processes may be more widespread than previously thought, and could influence future glacier behaviour in terms of retreat rate and ice flow velocity.

## 2 Study area

The Hardangerjøkulen icefield (60°55′ N, 7°43′ E) is situated in southern central Norway between a dry continental environment to the east, and the wet maritime climate of the Norwegian coast 150 km to the west (Fig. 1). The icefield covers 72.46 km² and ranges in altitude from 1020 to 1865 m a.s.l. (Andreassen and Winsvold, 2012). The icefield is located on a mountain plateau with an characteristic ice thickness of ~150 m, with ten minor and three major drainage basins where ice

thickness is up to 385 m (Fig. 2a-c) (Andreassen et al., 2015). Midtdalsbreen is sourced from a major north-eastern basin, which extends back to the ice divide with the south western outlet glacier Rembesdalskåka (Fig. 1b). Midtdalsbreen covers an area of 6.80 km² (Andreassen and Winsvold, 2012) and is ~4.8 km long, descending from 1862 to 1440 m a.s.l. The general lower limit of permafrost in this area is estimated to be 1550 m a.s.l. (Etzelmüller et al., 2003) but DC-resistivity soundings at 1450 m a.s.l. (Etzelmüller et al., 1998) and thermistor measurements of cold-ice (<0° C) at the glacier front of Midtdalsbreen (Hagen, 1978; Liestøl and Sollid, 1980; Etzelmüller et al., 1998) indicate permafrost at lower elevations, at least at snow-free sites (Lilleøren et al., 2013; Gisnås et al., 2014).

Maximum ice thickness at Midtdalsbreen is 280 m in the accumulation area, although the majority of the ablation area is < 120 m thick (Fig. 2c) (Willis et al., 2012; Andreassen et al., 2015). Ice flow velocity was measured as 31 m/yr in 2005-2006 (Giesen, 2009) on the relatively flat lower section of the glacier, but is likely higher in the heavily crevassed area where the glacier descends from the plateau, though no direct measurements are available here. Mass balance was +1.3 m w.e. in 2000 and -0.64 m w.e. in 2001 which equated to an upglacier shift in the equilibrium-line altitude from 1500 m to 1785 m (Krantz, 2002). Length variation measurements at Midtdalsbreen have been taken annually in the ablation season since 1982, close to the middle of the glacier snout near the centre flowline (Fig. 3a and b) (Nesje et al., 2008). Andersen and Sollid (1971) reconstructed the glacier's extent for the years 1934, 1955, 1959 and 1960 to 1968 using photographs and field observations.

In the majority of the study area, the LIA maximum extent is ~2 km NE of the 2014 glacier front. The LIA limit is defined by a distinct boundary between fluted and non-fluted terrain and by fragments of a terminal and lateral moraine that is best preserved to the NW of the current glacier front (Andersen and Sollid, 1971; Reinardy et al., 2013). Moraine fragments, flutes, and lichenometric data provide evidence of steady retreat of Midtdalsbreen from its position at the LIA moraine until its position during the 1930s, by which time the ice front had retreated ~0.5 km (Andersen and Bjørknes, 1978). Moraines formed after 1934 are limited, but glacifluvial terraces were deposited (Andersen and Sollid, 1971). Andersen and Sollid (1971) interpreted the lack of moraines during the 1930s and 1940s as indicative of rapid retreat. This may relate to the significant climatic shift that was observed in the latter part of the 1920s in western Norway, when average annual temperatures increased by 0.7°C and notably warm spring-summers occurred in 1930, 1933, 1937 and 1947 (Nordli et al., 2003). While no geomorphic record of moraines deposited during this time remain, flutes are preserved along the edges of a sandur in the glacier foreland. Between 1955 and 1968 only relatively minor (< 20 m) annual retreat took place (Andersen and Sollid, 1971). The glacier margin fluctuated annually by approximately ± 10 m during the 1980s but during the 1990s a minor (< 25 m/yr) readvance of Midtdalsbreen took place in response to several winters with high snowfall (Nesje et al., 2008). This readvance overrode (and thus destroyed) any moraines deposited during the previous decade (Fig. 3a and b). During more recent periods of warmer climate and rapid glacier retreat (up to 31 m in a single year) from 2001 to 2010, minor moraines continued to be deposited at the glacier snout (Reinardy et al., 2013). Retreat of the ice margin has continued from 2011 to 2018, with the exception of 2015 when the glacier advanced 19 m. By 2016 it had, however, retreated 21 m behind its 2015 position (Fig. 3a and b).

## 3 Methods

Ground-penetrating radar (GPR) methods have been widely used to image the beds and internal structures of glaciers, and are particularly useful for identifying englacial water inclusions within an otherwise frozen ice mass (Plewes and Hubbard, 2001). Englacial thermal boundaries have been mapped with GPR (Holmlund and Eriksson, 1989; Björnsson et al., 1996; Ødegård et al., 1997; Murray et al., 2000; Copland and Sharp, 2001; Pettersson et al., 2003; Gusmeroli et al., 2010; 2012). The technique is effective since liquid water within temperate ice causes radar energy to be backscattered, in contrast to cold-ice which has no water inclusions and therefore appears transparent (Gusmeroli et al., 2012). Water inclusions scatter energy because they represent a contrast in dielectric permittivity, and thereby in GPR wave speed. The wave speed in cold-ice is typically assumed to be ~0.17 m/ns (Bælum and Benn, 2011), and can be slowed by ~4% with the introduction of 1.4% liquid water (Endres et al., 2009). Regions of temperate ice can therefore be mapped by identifying regions of enhanced radar scattering, with the cold-temperate transition surface (CTS) inferred at the transition from transparent to chaotic regimes.

Forty-three kilometres of GPR line data were collected during April 2014, primarily around the snout of Midtdalsbreen (Fig. 4a-c), when the glacier was snow covered. Dense coverage at the glacier snout consists of grids with 5 m line spacing, facilitating 3-D imaging across the whole snout (Fig. 4c). Away from the marginal zone, coverage is limited to sparse profiles, and a series of isolated acquisitions around the Midtdalsbreen/Hardangerjøkulen ice divide and two nunataks on the eastern and western side of the glacier (Figs. 4a and b).

All GPR data were acquired with a Malå Geosciences Rough Terrain Antenna (RTA) of 50 MHz centre-frequency. The RTA is a bistatic system, in which transmitting and receiving antennas are mounted in parallel end-fire orientation at 3.5 m separation. A GPR trace was recorded every 0.5 s. GPS positions were recorded by a backpack-mounted antenna, offering +/- 4 m positional accuracy; a GPS position was recorded every 1 s. Throughout, the midpoint of the RTA was 6.2 m behind the operator (motivating a positional correction in later processing). Acquisition of the 3-D grids was conducted on foot, with each line separated by ~5 m and an along-profile trace sampling interval of $0.47 \pm 0.12$ m. For the longer lines, the RTA was towed behind a snowmobile at somewhat higher speed, resulting in a trace sampling interval of $0.78 \pm 0.18$ m. For 50 MHz energy propagating in ice (velocity ~ 0.17 m/ns), with a wavelength of 3.4 m, our along-line sampling interval satisfies the quarter-wavelength resolution criterion of Grasmueck et al. (2005), but grids may be spatially aliased in the cross-profile direction. Our limit of vertical resolution is ~ 0.85 m, although our estimates of ice thickness are likely more accurate than this given the typically clear first-break arrival times.

Data pre-processing was conducted in Sandmeier ReflexW and comprised bandpass filtering (corner frequencies of 15-30-70-140 MHz) and static corrections to synchronise first-break travel-times to 11.7 ns (the travel-time of the direct airwave across the 3.5 m of antenna offset). From these data, regions of cold and temperate ice were identified and geo-located using GPS

positions corrected for the offset between the GPS receiver and the midpoint of the RTA system (in the direction of survey azimuth along each profile). The CTS was identified from unmigrated data since it is most effectively defined when diffraction hyperbolae are present (e.g., Fig. 5a). Migration was applied thereafter to establish the geometries of the snow cover and glacier bed. To facilitate migration, the trace spacing in all profiles was regularised in Mathworks Matlab; traces in the marginal

grids were also interpolated onto regular grids with sample dimensions of $0.2 \times 5$ m. Data were then passed back into ReflexW and Kirchhoff migrated with a constant ice velocity of 0.17 m/ns (Bælum and Benn, 2011). A variable velocity field was unavailable since common midpoint (CMP) acquisition is not feasible with the RTA. Nonetheless, the majority of diffraction hyperbolae in our dataset were appropriately characterised by this velocity and became well-focused. Travel-time picks of the glacier surface (i.e., the base of the snow cover), glacier bed and CTS were gridded using a minimum curvature algorithm in

Golden Software Surfer. Grids targeting the glacier snout are gridded at $2 \times 1$ m resolution, whereas those made for the whole glacier have resolution $20 \times 20$ m. Depth conversions assumed a snow velocity of 0.23 m/ns (Holbrook et al., 2016) and an ice velocity of 0.17 m/ns. Considering typical variations about these ranges, we estimate that depth conversions are accurate to $\pm 2$ m, which is small with respect to the large-scale variation we observed.

Geomorphological mapping was carried out updating and expanding on previous geomorphological maps from the area by Sollid and Bjørkenes (1978) and Reinardy et al. (2013). Mapping was compiled from field observations during three field campaigns in 2013, 2016 and 2017 alongside 0.25 m resolution aerial photographs taken in 2013 (acquired from https://www.norgeibilder.no/) using protocols outlined in Chandler et al. (2018). Sedimentological analysis, following procedures outlined by Evans and Benn (2004), was also used alongside the geomorphological mapping to provide additional

information on sediment transport and on the processes leading to landform genesis. Exposures through flutes and ice-cored ridges on the glacier snout where a meltwater stream exposed debris-rich ice layers were logged and are briefly described below. Debris-rich ice was exposed at numerous other location on the south-eastern glacier snout and a covering of supraglacial debris was also mapped and described from this location. Occasionally it was also possible to see debris melting out of debris-rich glacial septa at the ice surface. A number of sections were logged through de-iced hummocky terrain in proximity to the

south-eastern glacier margin and the details summarised below.

## 4.1 Results – Geomorphology and sedimentology of the foreland

The most widespread glacial landform across the foreland of Midtdalsbreen are flutes, sets of which can be traced from the LIA limit to the current glacier margin where they emerge from under the glacier snout (Fig. 6). Frozen flute ridges are normally found incorporated into the basal ice layer at the glacier snout. Flutes at the ice front tend to be <60 cm in height, up

to 40 m in length and <1 m wide and composed of diamicton. They are observed in particularly high concentrations in the central foreland, lower concentrations at the NW glacier margin, and are absent from the south-eastern glacier margin (Fig. 6). The minor moraines at the NW glacier margin have previously been described by Reinardy et al. (2013). The south-eastern margin contains several landforms and associated sediments that relate to a process-form continuum, involving the

concentration of supraglacial debris, controlled moraine formation and de-icing of controlled moraines into hummocky moraine (sensu Evans, 2009) (Figs. 6 and 7). The south-eastern glacier snout is covered by supraglacial debris, normally <20 cm thick, massive and primarily consisting of sands and gravels although some coarser debris from the valley sides is also present. In particular, melt-out of the bedload of former drainage channels on the glacier surface is continuously being reworked by supraglacial meltwater and gravitational sliding to form an increasingly thick and continuous debris cover at the glacier margin, as described from similar debris-covered cold-ice margins (e.g. Lukas et al., 2005). Ice-cored debris ridges, in some cases emanating from englacial debris septa, also occur along the glacier snout and margin and are interpreted as controlled moraines (Figs. 6 and 7). A basal ice layer was previously described along the glacier sole of the south-eastern glacier margin consisting of layers of debris-rich ice between 0.3 and 0.5 m thick that incorporated both fine sediment (sands and clayey silts) and occasional bedrock clasts (phyllites) (Reinardy et al., 2013). This limited exposure also showed that the dipping basal ice layer was stratified and incorporated subglacial material entrained from the frozen substrate that was then elevated to an englacial and likely supraglacial position. These elevated, debris-rich basal ice layers likely correspond to the debris emanating from englacial septa on the glacier snout forming controlled moraines. Evans (2009) notes that polythermal conditions are crucial to the concentration of supraglacial debris and formation of controlled moraines on glacier snouts via processes that are most effective at the glacier-permafrost interface. This is because sub-zero temperatures below the glacier snout and margin would favour adfreezing and subsequent entrainment of sediments into the basal ice layer (Weertmann, 1961; Etzelmüller et al., 2003; Etzelmüller and Hagen, 2005; Myhra et al., 2017). The lower limit of permafrost in this area corresponds to approximately the glacier front of Midtdalsbreen at 1440 m a. s. l. (Etzelmüller et al., 1998; 2003; Lilleøren et al., 2013; Gisnås et al., 2014).

The controlled moraines ridges, both on the present ice margin and in the immediate foreland, vary greatly in size relating to level of degradation but are generally <5 m high and <2 m in width. The controlled moraines primarily consist of well-sorted, stratified sand and gravel. Debris flows are ubiquitous at the south-eastern glacier margin and in the immediate foreland, and along with meltwater stream downcutting, expose underlying ice which then allows rapid melting, in turn leading to further debris flows. Thus, due to self-reinforcing degradation, the controlled moraine surface consist of a range of morphologies, from ridges with well-defined crestlines to conical-shaped "dirt cones" (Fig. 7). Further away from the present ice margin, the zone of hummocky moraine is interpreted as a direct product of melted out controlled moraine due to similar sedimentological characteristics when compared to the controlled moraines. The de-iced hummocky moraine ridges are also relatively linear in form although, as with the controlled moraines, levels of degradation are highly variable. The main sedimentological difference occurs due to melt-out of the ice core. The well-sorted sand and gravel beds dip from 18-25° where ice-core support has been removed. Flatter areas of glaciofluvial and/or glaciolacustrine sediments are present between and surrounding the area covered by de-iced hummocky moraine. It is possible that these sediments along with the hummocky moraine may still be underlain by some areas of remnant dead-ice and/or permafrost.

## 4.2 Distribution of cold-ice

Two distinct responses, transparent and chaotic, can be seen in the GPR profiles (Figs. 5a-c and 8a-d; profile numbers hereafter have a prefix "PR"), corresponding respectively to ice with and without inclusions of liquid-water. Thus, this allows delineation of areas of cold-ice and temperate ice at the pressure melting point. In the majority of cases, the CTS is sharply defined (Figs. 5a, b; 8a, b and d). The glacier bed appears as a highly reflective horizon until 500 m from the snout, where ice thickness exceeds 70 m and signal-to-noise ratio becomes poor owing to a combination of signal attenuation and/or interference with scattered energy (Fig. 8a and b). At the snow-covered glacier snout it was also possible to estimate where the glacier margin was located from GPR data and compare this to measured length variation data carried out in the late summer of 2014 (Andreassen et al., 2015). The distribution and thickness of cold-ice in the ablation area is shown in Figure 6 interpreted from the GPR data.

Results from the GPR data indicated that all surveyed areas had a cold-ice upper layer 8 m thick probably caused by cold conditions during the winter and then insulated by snow-covered during spring (April) when collection of the GPR data took place (Figs. 5a-c and 8a). Further surveys would need to be carried out at the end of the summer melt season to establish if this surface cold-ice layer is seasonal. Areas of the glacier that have excess cold-ice (>8 m thick), or that have cold-ice extending to the glacier bed, are here termed cold-ice zones (CIZ) (Fig. 6). Three CIZs were found in the ablation area (CIZ1-3, Figs. 5a-c; 8a-d) and two CIZs in the accumulation area in proximity to the western and eastern nunataks (CIZ4-5, Figs. 4a and 9). CIZ1 is a 40-50 m wide corridor around the glacier snout where ice thickness is ≤10 m (Fig. 5a and b). The GPR data indicate that CIZ1 contains cold-ice that extends to the glacier bed at the glacier snout (Fig. 5a-c). CIZ2 and CIZ3 are up to 15 m thick and approximately 100 and 200 m wide respectively (Figs.6 and 8a, b). CIZ2 is located on the eastern margin of the glacier in the ablation area. Ice thickness here is < 40 m and thus CIZ2 may extend to the glacier bed causing basal freeze-on, though further GPR surveys are necessary to confirm this assumption (Fig. 8b and c). CIZ3 occurs approximately half way up the length of the glacier and appears to extend across its full width (Fig. 6). Our GRP wavelet is not able to image the glacier bed at this point (Fig. 8a) but previous studies indicate ice thickness here to be 50-100 m (Fig. 2c) (Willis et al., 2012; Andreassen et al., 2015). We therefore assume that CIZ3 does not extend to the glacier bed except at its eastern margin (Fig. 8d). Lastly, two nunataks in the western and eastern accumulation area of the glacier have even larger cold margins, CIZ4 and CIZ5 respectively, which extend 100 m out from the nunataks and extend to the glacier bed at ~ 30 m depth (Figs. 4a and 9).

## 5.1 Discussion – Cold-ice zones in the ablation area

Photographic monitoring of the ice front by Giesen (2009) during 2005-2008 shows that the glacier snout remains snow covered until late summer (end of July beginning August) while ablation areas further upglacier are snow free by the start of July (Fig. 10). Late-lying seasonal snow patches (i.e. present until the beginning of August) in this location would promote formation of cold-based ice by insulating the ice surface from warming during the start of the summer melt season. Giesen (2009) relates this pattern of snow melt to the greater snow depths caused by drifting at the relatively steep glacier snout.

Similar to CIZ1, both CIZ2 and CIZ3 in the ablation area correspond to parts of the glacier that experience snow drifting facilitated by glacier topography and correspond to areas of late-lying seasonal snow cover in multiple years (Fig. 11a-c). Aerial photos of Midtdalsbreen were available for 2004, 2007, 2010, 2013, 2014 and 2017 and all show late-lying snow patches in approximately the same position until at least the beginning of August, although in some years, such as 2007, these snow patches remained until mid-August (Fig. 11b). This suggests the late-lying seasonal snow patches have remained stable during at least the summer melt seasons of 2004, 2007, 2010, 2013, 2014 and 2017, implying that the thermal regime may also have remained relatively stable during the last decade (Fig. 11a-c). By the beginning of September most of these snow patches have melted away (Fig. 4a). Snow depths above 0.6–0.8 m have been found to effectively insulate the ground from the atmosphere (Luetschg et al., 2008). Gisnås et al., (2014) measured spatially variable ground temperatures over small areas (<500 m$^2$) near Midtdalsbreen linked to snow redistribution caused by wind drift which in turn creates a pattern of different snow depths. When modelling permafrost distribution, Myhra et al., (2017) found that varying snow depth could have a significant influence on surface ground temperature. Thus, it is very likely that the distribution and/or redistribution on snow cover on the surface of Midtdalsbreen also has a strong influence on the underlying thermal regime of glacier ice. Areas of late-lying snow and reduced ablation have also previously also been linked to thickening of a cold surface layer by insulating the ice from warmer air temperatures on Storglaciären in northern Sweden (Gusmeroli et al., 2012).

## 5.2 Sediment transport and depositional processes at the glacier snout

The distribution of cold and temperate ice has been recognised as an important factor for the production, incorporation, transport and deposition of glacial debris (e.g. Weertman, 1961; Etzelmüller and Hagen, 2005; Evans, 2009). Ice-cored moraines and debris covered dead-ice areas around a glacier margin regulate sediment transfer since debris release depends on the removal of material protecting the ice-core, and this process happens during the summer melting season independent of glacier activity (Etzelmüller and Hagen, 2005). The identification of a cold-based snout and a number of other CIZs in both the ablation and accumulation areas of Midtdalsbreen supports the earlier interpretation of a cold-based thermal regime at the glacier snout made from detailed examination of the sedimentary architecture of the moraines (Fig. 5a-c) (Reinardy et al., 2013). However, while Reinardy et al. (2013) estimated that the cold-based snout extended around 10 m up-glacier from the glacier margin at the end of the summer, this study indicates that from the glacier margin at least 40-50 m of the glacier snout contains cold-ice (CIZ1) extending down to the ice-bed interface where it is probably frozen to the bed at least during the spring (Fig. 5a-c). This is likely facilitated by continued thinning of the glacier snout, allowing the winter cooling to penetrate through to a larger area of the bed.

The dense GPR survey grids over the glacier snout provide a detailed imaging of the CTS (Figs. 4c; 5a-c). The CTS favours adfreezing of sediments into the basal ice layer and their transport to englacial and supraglacial locations (Etzelmüller and Hagen, 2005). This likely governs sediment supply to the margin of Midtdalsbreen, which in turn exerts direct control on moraine formation. Water flowing at the bed from warm to cold thermal zones can freeze to the glacier sole allowing adhesion

or incorporation of sediment (Knight, 1997). Weertman (1961) described debris entrainment occurring when the freezing isotherm passes downwards into the substrate and subsequent thickening of the basal layer by sequential addition of layers of new ice at the bed by freeze-on. This process was investigated further by Dobiński et al. (2017) at Storglaciären where the CTS connects with the base of the permafrost following the freezing isotherm underneath the glacier separating frozen and

5 unfrozen subglacial strata. Thus, the CTS forms an environmental continuum with its equivalent boundary in the periglacial environment corresponding to the base of the permafrost (Dobiński et al., 2017). Etzelmüller and Hagen (2005) previously modelled a similar thermal regime at Midtdalsbreen where the CTS at the glacier bed is linked to the base of the permafrost that extends down into the subglacial substrate below the glacier snout. In this study we show possible evidence of this freezing isotherm defining the base of the permafrost under the snout of Midtdalsbreen and extending downwards into the subglacial

strata (labelled "BP" in Fig. 5b). Distinct dipping reflectors in the GPR data can be seen below the glacier bed in PR2, tentatively interpreted as the base of the permafrost. Thus, it is likely that frozen till underlies the cold-ice glacier snout and underlying the frozen till is either unfrozen till or bedrock. Lower-frequency antennas, or alternate geophysical approaches (Killingbeck et al., 2018), could confirm this interpretation.

Much of the foreland of Midtdalsbreen is fluted (Fig. 6). Flutes have previously been linked to glaciers with extensive cold-based margins (e.g. Gordon et al., 1992; Roberson et al., 2011). In addition, flutes are described as having a limited preservation potential (Benn and Evans, 2008), but within the timeframe extending back to the LIA, they are relatively well preserved on the foreland of Midtdalsbreen. Our observation that flutes at Midtdalsbreen are frozen to the glacier bed at the glacier snout suggests that flutes either actively form beneath cold-based ice, or that they form beneath warm-based ice, possibly by

streaming of basal ice around subglacial obstacles (e.g. Gordon et al., 1992), and are subsequently frozen-on. For polythermal glaciers, Eklund and Hart (1996) suggest flutes form beneath warm-based ice and are then frozen as the CTS passed over them. Thus, for flutes to form at polythermal glaciers by this mechanism, migration of the CTS is required, indicating that flutes are produced only at retreating and thinning glaciers as is the case with Midtdalsbreen.

While the north-western part of the glacier foreland contains minor moraines, the south-eastern part of the glacier margin is debris-covered, resulting in an uneven ice surface formed of debris-covered ridges interpreted as controlled moraines (Figs. 6 and 7). Many of the controlled moraines are located where debris septa melt out at the ice surface as described by Evans (2009) (Fig. 7). This zone of controlled moraines is in stark contrast with the area of minor (possibly annual) moraines in the north-western part of the foreland, indicating a range of distinct processes of sediment delivery to the ice margin across the

Midtdalsbreen foreland (Fig. 6). However, both sets of moraines are considered as indicators of permafrost conditions at the ice margin. This supports previous interpretations of permafrost at the margin of Midtdalsbreen (Etzelmüller et al., 2003; Etzelmüller and Hagen, 2005). De-iced hummocky moraines are formed once the ice-core of the controlled moraines melts out at the glacier margin or on the foreland. Etzelmüller and Hagen (2005) note that in permafrost environments, a thick debris cover may preserve underlying ice over long time periods. However, water may locally remove the loose cover material, or

mass movement processes may expose ice-cores, accelerating ice-core decay under permafrost conditions, and resulting in the formation of hummocky terrain (Etzelmüller and Hagen, 2005).

In 2013 and 2017, debris was observed within debris-rich glacial septa and crevasses emerging from an englacial source about 200 m up glacier from the then ice margin. This emerging debris contributed directly to the zone of supraglacial debris at the south-eastern glacier margin (Fig. 7b). Previous studies have shown that debris-rich basal ice in polythermal glaciers can be elevated to englacial and supraglacial positions via compressional glacitectonics at the CTS, where temperate ice is being thrusted over cold stagnant ice or as a result of recumbent folding within the glacier (Glasser et al., 2003). CIZ2 would have provided these conditions for the formation of controlled moraines at Midtdalsbreen. Evidence of debris elevated from an englacial (and likely subglacial) position to the ice surface can been seen in the GPR data directly up glacier from CIZ2 (Fig. 8b). The GPR data also indicate what appears to be a small mound on the surface of the glacier that occurs where the interpreted englacial debris layer intersects the ice surface. This mound may well be a small controlled moraine buried by snow at the time of surveying (April 2014). However, an alternative explanation of debris elevation at a glacier with a frozen margin is proposed by Moore et al., (2011). They presented model and field evidence from the terminus of Storglaciären, Sweden, showing that the cold margin provides limited resistance to flow from up-glacier. Instead they found that sediment elevation from a subglacial to supraglacial position was primarily influenced by a subglacial "bump" at the glacier bed in the vicinity of the CTS. A similar topographical bump, <2 m high, is seen in GPR PR3 on the extreme south-eastern margin of Midtdalsbreen (Fig. 5c). Directly downstream from the bump are englacial GPR reflectors that may correspond to the elevation of debris. A similar englacial GPR response was observed for elevated debris layers at the snout of Kongsvegen, Svalbard (Murray and Booth, 2009). However, the subglacial bump shown in PR3 (Fig. 5c) does not extend across the whole width of the south-eastern snout area, thus, both subglacial topography and/or longitudinal stress gradients at the CTS may elevate subglacial and englacial debris to the ice surface. It should be noted that following the study of Moore et al., (2011), Dobiński et al., (2017) still describe some areas of Storglaciären where, as a results of compressive motion, basal debris is elevated to a supraglacial position. Both surface velocity measurements and boreholes in the vicinity of the subglacial bump would be needed to further investigate the relative influence of the mechanism(s) of debris elevation processes at the snout of Midtdalsbreen.

**5.3 Cold-ice zones in the accumulation area**

Several studies have indicated partly frozen ice sheet beds (Bentley et al., 1998; Gades et al., 2000) as well as temperate glaciers or ice caps and icefields with a frozen marginal zone (Waller et al., 2012 and references therein). Here, we show that nunataks in the accumulation area of temperate glaciers can also be surrounded by a cold-ice zone that extends to depths greater than the comparatively thin surface cold layer. For Midtdalsbreen, these zones are relatively large, extending up to 100 m out from the nunatak margins and down to the glacier bed (CIZ4 and CIZ5, Fig. 9). Both nunataks are at altitudes >1800 m a.s.l., well above the lower limit of permafrost for this area (Etzelmüller, 2003). The distribution of permafrost within the nunataks is likely to have a significant influence on the surrounding glacier thermal regime. Modelling suggests that heat fluxes below

glaciers can be influenced by neighbouring exposed rock walls (Myhra, et al., 2017). At these locations, permafrost can extend horizontally and under glacier ice, even at locations where surface temperatures would suggest otherwise (Noetzli et al., 2007; Myhra et al., 2017). It has also been demonstrated that glacier retreat due to negative surface mass balance would in some cases favour the development of near-surface permafrost from increased exposure of steep snow-free rock walls (Kneisel et al., 2000) while irregularities on the surface of nunataks can modify ground temperatures and promote local permafrost occurrence (Noetzli et al., 2007). The area around the western nunatak (CIZ4) thinned by approximately 2 m from 1961 until 1995 (Andreassen and Elvehøy, 2001) but thinning is likely to have continued to the present day considering the trends of neighbouring glaciers (Andreassen et al., 2016). In contrast, the eastern nunatak (CIZ5) showed some of the highest rates of ice thickening across the whole Hardangerjøkulen icefield from 1961 until 1995 (Andreassen and Elvehøy, 2001). Therefore, we hypothesise that CIZ5 around the eastern nunatak may be a relatively recent feature, directly related to icefield thinning over the last two decades while the formation of CIZ4 may have started several decades earlier.

While the GPR data did not penetrate to the bed over the rest of the plateau, the presence of temperate ice at the surface of the thickest parts of the Hardangerjøkulen icefield suggests that these areas are entirely temperate. However, it is possible that additional parts of the icefield also are cold-based, for example close to the ice divide where ice thickness is < 50 m (Figs. 1b; 2a and b) or where additional nunataks are present. Further GPR investigations are needed to confirm this. As the icefield continues to thin (Andreassen et al., 2016), probably accompanied by a corresponding decrease in ice flow velocity, cold-ice and/or frozen bed conditions may become more widespread with future climate warming. Thus, it is likely that cold-ice zones may become more extensive over the icefield and upper accumulation areas but conversely possibly less extensive in the ablation area where late-lying seasonal snow patches are likely to decrease in number and size in future decades.

## 5.4 Implications and wider significance of cold-ice

Extensive cold-ice zones could affect ice flow by means of increased shear between temperate and cold-ice zones within the glacier, and/or higher drag along cold-based margins (Åkesson, 2014). Whether the margins of Midtdalsbreen are underlain by unlithified sediment and/or bedrock likely determines the hypothesised changing stress regime and effect on glacier dynamics. A till-covered bed likely influences basal motion, as pointed out by Meyer et al. (2018). They argue that depending on local thermodynamics, ice can infiltrate subglacial sediments and thereby control the bed strength and basal sliding. Annual average surface ice velocity was measured to 33 m/yr at the Equilibrium Line Altitude of Midtdalsbreen during 2005-2007 (Giesen, 2009), but flow is likely markedly slower in areas with ice frozen to its bed, such as CIZ4 and CIZ5 (Fig. 9). However, absence of more spatially extensive velocity data at Midtdalsbreen currently makes a detailed dynamical assessment difficult. Moore et al. (2011) measured only very low longitudinal compressive stresses at the frozen margin of the polythermal Storglaciären in northern Sweden. They suggested that these low stresses result from weak unfrozen till beneath the freezing isotherm, which allows basal motion beneath cold-ice all the way to the glacier margin. As discussed earlier, while there is some evidence for the freezing isotherm extending down into the subglacial strata at the south-eastern margin of Midtdalsbreen,

it is likely that the entire margin of Midtdalsbreen and surrounding nunataks are underlain by a mixture of bedrock and till both frozen and unfrozen.

Cold-ice is also more viscous ("stiffer") and deforms less readily than temperate ice (Cuffey and Paterson, 2010, p. 12). Therefore, cold-ice zones in predominantly temperate glaciers may act as a negative feedback on ice flow in a warming climate. Elevation changes measured on Midtdalsbreen between 1961 and 1995 showed that it had thinned by 5-10 m in the ablation area (Andreassen and Elvehøy, 2000). If cold-ice zones become more widespread due to glacier thinning, as suggested by the likely recently formed cold-ice zones in the accumulation area at Midtdalsbreen, we may see a future deceleration and further preferential thinning of previously temperate glaciers like Midtdalsbreen. Changing ice properties and its impact on large scale flow is also relevant on the ice sheet scale; variable ice rheology has been suggested to have caused deceleration of the Greenland Ice Sheet during the Holocene (MacGregor et al., 2016). However, continued research including measurements of mass balance and surface ice flow velocity are needed to determine if glacier slowdown is occurring at Midtdalsbreen. Glacier deceleration and an associated reduced erosive power of the glacier would also have significant implications for both the sediments and landforms deposited within the glacier foreland (e.g. Koppes et al., 2015). For example, changes to glacier basal thermal regime and resulting altered glacier dynamics may bias glacier reconstructions based on proglacial lake sediments (Åkesson et al., 2017), which assume a relationship between eroded glacier flour and glacier variability (e.g. Hallet et al., 1996). In addition, disintegration and melt-out of ice-cored moraines and release of supraglacial debris likely cause rapid pulses of sediment release stored along cold-based glacier margins (Etzelmüller, 2000). In areas where extensive cold-ice has been interpreted to have been frozen to the bed, i.e. cold-based plateau icefields or ice caps and outlet glaciers, the characteristic geomorphic imprint has previously been highlighted (e.g. Dyke, 1993; Rea and Evans, 2003; Evans, 2010; Pearce et al., 2014; Boston et al., 2015; Boston and Lukas, 2017). However, considering that both the minor and controlled moraines at Midtdalsbreen are, or were ice-cored, their preservation potential is very limited (e.g. Lukas et al., 2005; Reinardy et al., 2013). Thus, cold-based processes may also be far more widespread in the temperate palaeoglacier record than currently accounted for.

This study has wider significance in terms of glacier thermal regime and how thermal properties relate to the glacial geomorphological record. Firstly, glaciers that are defined as temperate may contain considerable areas of cold-ice. While some studies have previously shown polythermal glaciers with a frozen margin (e.g. Pettersson et al., 2007), here we show that in addition to a cold-ice margin during spring within a temperate glacier, parts of the upper accumulation area can also contain cold-ice, if ice is relatively thin, for example in proximity to nunataks or adjacent rockwalls. Both these characteristics apply to many plateau-type ice caps, icefields, and outlet glaciers. Thus, it is likely that cold-ice processes, including subglacial freeze-on processes, could be more widespread than previously thought. Based on numerical modelling of ice flow and surface mass balance, Giesen and Oerlemans (2010) suggest that the Hardangerjøkulen icefield could disappear by year 2100. However, it is difficult to predict the future distribution of cold-ice for both glaciers and icefields. Thinning of cold-ice layers

has been linked to regional climate change (Gusmeroli et al., 2012), and results from this study suggest that CIZ's are likely to form below late-lying seasonal snow patches on the glacier surface, which will presumably also shrink and disappear in future warming climates. However, ice thinning, possible ice deceleration and retreat of glacierized areas to altitudes where permafrost becomes more prevalent, may increase the presence of cold-ice and therefore cold-based processes at least for a

limited period during overall retreat (Rippin et al., 2011). There is a growing body of research focusing on Svalbard (e.g. Lovell et al., 2015) that suggests that historical and ongoing climatic changes have resulted in a thinning and deceleration of glaciers. This has in turn resulted in a reduction in the extent of warm-based ice and a progressive shift from polythermal to entirely cold-based thermal regimes. Based on our observations at Midtdalsbreen, we suggest that this counterintuitive cold-ice expansion in a warming climate may also be taking place in more temperate environments, with primarily temperate

glaciers becoming polythermal. Spatial and especially temporal variability in ice rheology and particularly thermal regime is currently poorly represented in numerical ice flow models. Many modelling studies currently do not account for changes to the thermal regime in their predictions, nor the potential subsequent effects on ice dynamics discussed above. Climate forcing is the main control on mass balance of land-terminating glaciers, icefields and ice caps, but the heterogeneous thermal conditions found at Midtdalsbreen and potential presence and future cold-ice expansion at other similar temperate glaciers,

calls for improved model assessments of evolution of thermal conditions and associated impact on glacier dynamics, mass balance and the palaeorecord.

## 6 Conclusions

Midtdalsbreen is a temperate glacier but previous studies identified cold-based ice at the snout which is also underlain by

permafrost. This study presents the first systematic mapping of the glacier using GPR, which indicated that cold-ice was far more widespread than previously assumed. Within the ablation area, not only was cold-ice measured at the snout (CIZ1) but also the lateral margins of the glacier and in two distinct bands across it (CIZ2 and CIZ3). CIZ1-3 correspond to parts of the glacier surface covered by late-lying seasonal snow patches that occur in approximately the same location during multiple years. Extensive cold-ice was also identified in the vicinity of two nunataks in the upper accumulation area (CIZ4 and CIZ5).

The south-eastern glacier margin contains several landforms and associated sediments that relate to a process-form continuum caused by concentrated supraglacial debris, leading to the formation of controlled moraines and subsequent de-icing of these controlled moraines into hummocky moraine. GPR data from the glacier snout indicate that accumulation of debris into the basal ice layer and subsequent elevation to a supraglacial position may be influenced by the CTS, a subglacial bump and the

location of the freezing isotherm within the underlying substrate. With predicted continued warming in future decades, areas of cold-ice may decrease due to reduced or disappearing late-lying seasonal snow patches. Conversely, continued thinning of the icefield and its outlet glaciers and probable reduction in ice flow velocity, as well as glacier retreat to higher altitudes, may promote more widespread cold-ice and processes such as basal freeze-on in the short to medium term at least. The most striking

implication of this study is the possibility that temperate environments with primarily temperate glaciers could become polythermal, similar to recently reported high Arctic glaciers transitioning from polythermal to entirely cold-based.

*Author contribution.* B. Reinardy prepared the manuscript with contributions from all co-authors and all co-authors were also involved in the construction of figures and analysis of GPR and geological data sets. Fieldwork and collection of GPR data was carried out by B. Reinardy, A. Booth, A. Hughes and J. Bakke. Geomorphological mapping was carried out by B. Reinardy, C. Boston and D. Pearce. We also wish to thank Richard Waller, Bernd Etzelmüller and Wojciech Dobinski for insightful and helpful reviews and comments which greatly improved this manuscript.

*Competing interests.* The authors declare that there is no conflict of interest.

*Acknowledgements.* The research leading to these results received funding from INTERACT (grant agreement no. 262693 GIMMIC and PLATREAT) under the European Community's Seventh Framework Programme. We thank the staff at the Finse Alpine Research Centre for logistical support, particularly Erika Leslie, Torbjørn Ergon and Kjell Magne Tangen. We also wish to thank K. Melvold for providing ice thickness data and Peter Jansson for helpful feedback and discussion of ideas in this paper. Codes for implementing GPS corrections were developed by Helena Lewczynska (MSc Exploration Geophysics, University of Leeds).

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

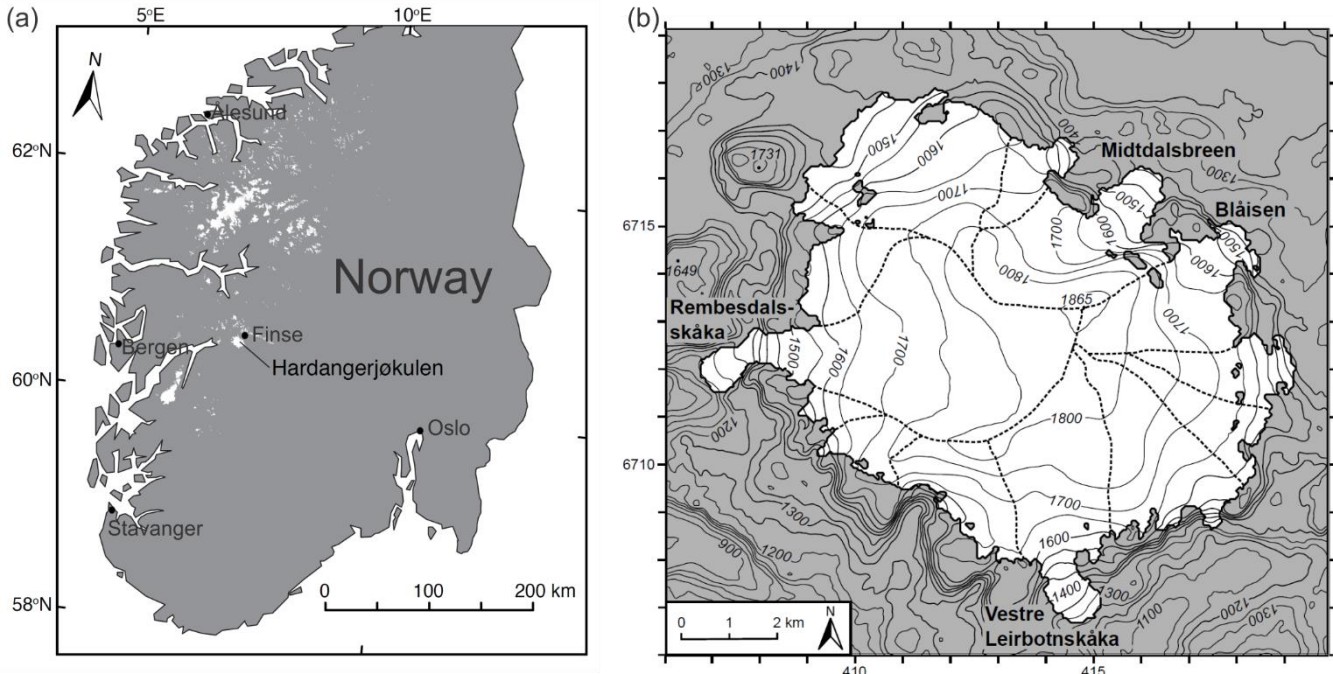

**Figure 1: (a) Map of south central Norway with major ice bodies indicated in white (modified from Giesen, 2009). (b) Hardangerjøkulen icefield created from a 1995 digital elevation model by Statens Kartverk (modified from Giesen, 2009; UTM zone 32 (EUREF89), 50 m contour interval. Dashed lines indicate the estimated extent of the drainage basins of outlet glaciers (Andreassen and Elvehøy, 2001).**

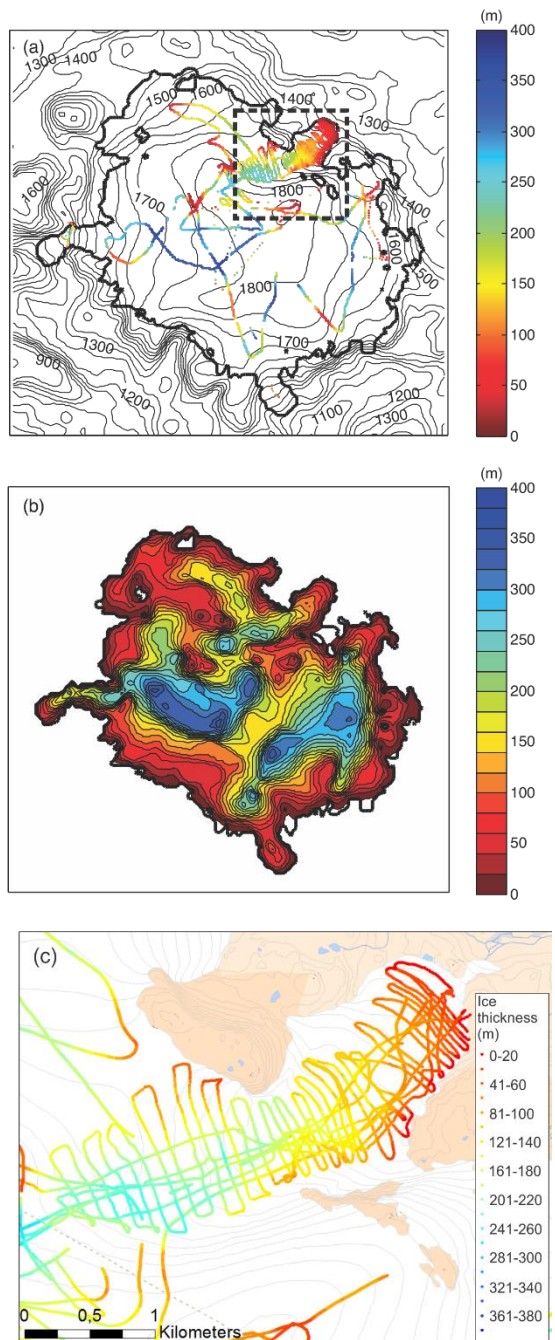

Figure 2: (a) Ice thickness data from studies 1963-2008. Black dashed box indicates location of (c). (b) Ice thickness calculated from the 1995 surface DEM and thickness measurements from studies 1963-2008. (c) Detailed ice thickness data from Midtdalsbreen. Figures from Giesen (2009) and data from K. Melvold, NVE.

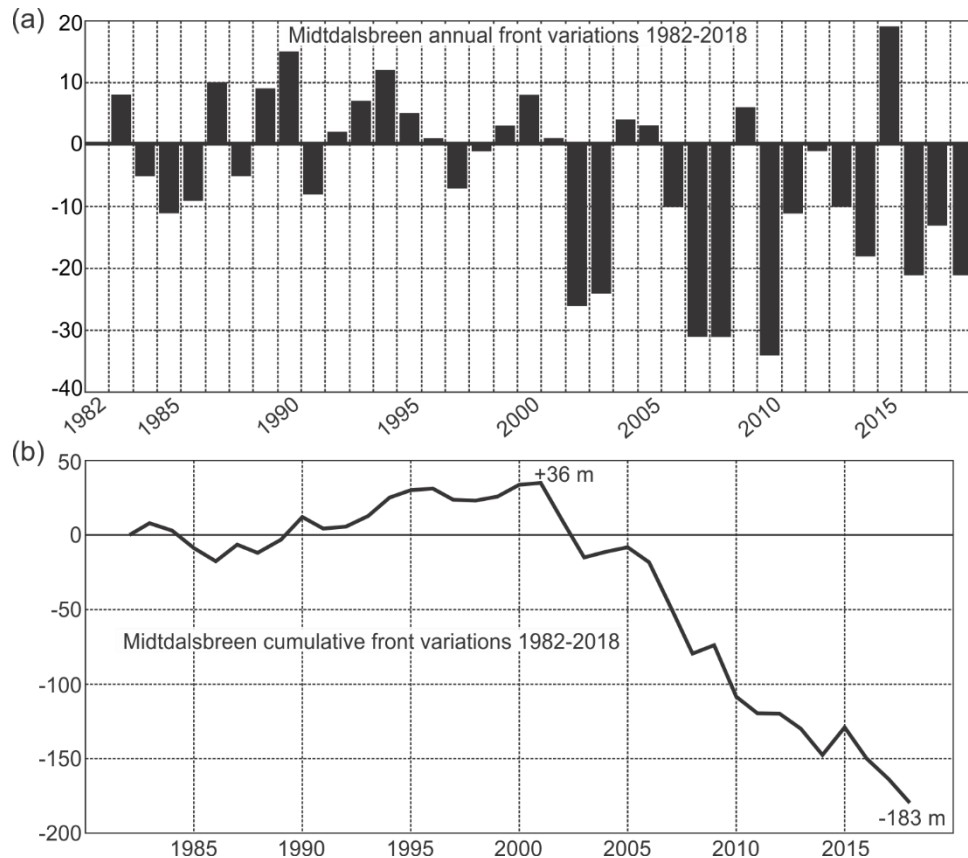

**Figure 3: (a) Midtdalsbreen annual front variations from 1982-2018 and (b) cumulative front variations from the same time period. Date collected and compiled by A. Nesje.**

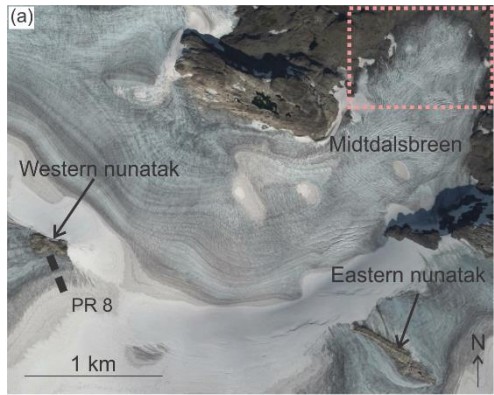

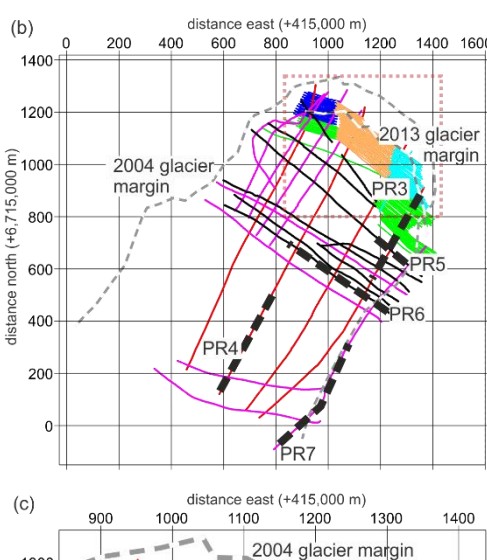

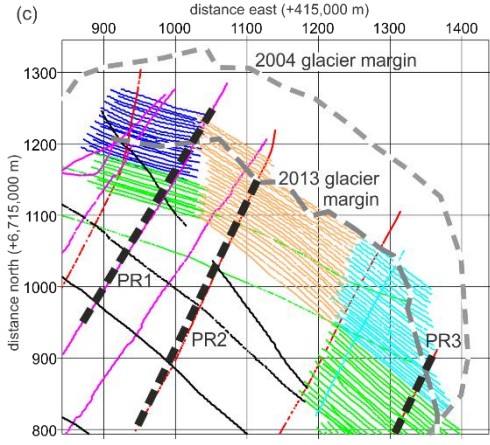

**Figure 4: (a)** Locations of western and eastern nunataks in the upper accumulation area of Midtdalsbreen (Google Earth imagery 04/09/2014). Note by this time in the melt season most snow patches have melted from the glacier surface. Dashed black line shows location of profile (PR) 8 (Fig. 9). Pink dashed box shows **(b)** locations of GPR surveys (coloured areas and solid lines) acquired on Midtdalsbreen 20-26/04/2014, with the 2004 and 2013 glacier margin (dashed grey) digitised from Google Earth. Thick black dashed line indicates locations of PR 3-7 (Figs. 5c; 8a-d). Pink dashed box shows **(c)**, enlarged window of dense surveys (coloured lines) on the glacier snout. Thick black dashed line indicates locations of PR 1-3 (Fig. 5a-c).

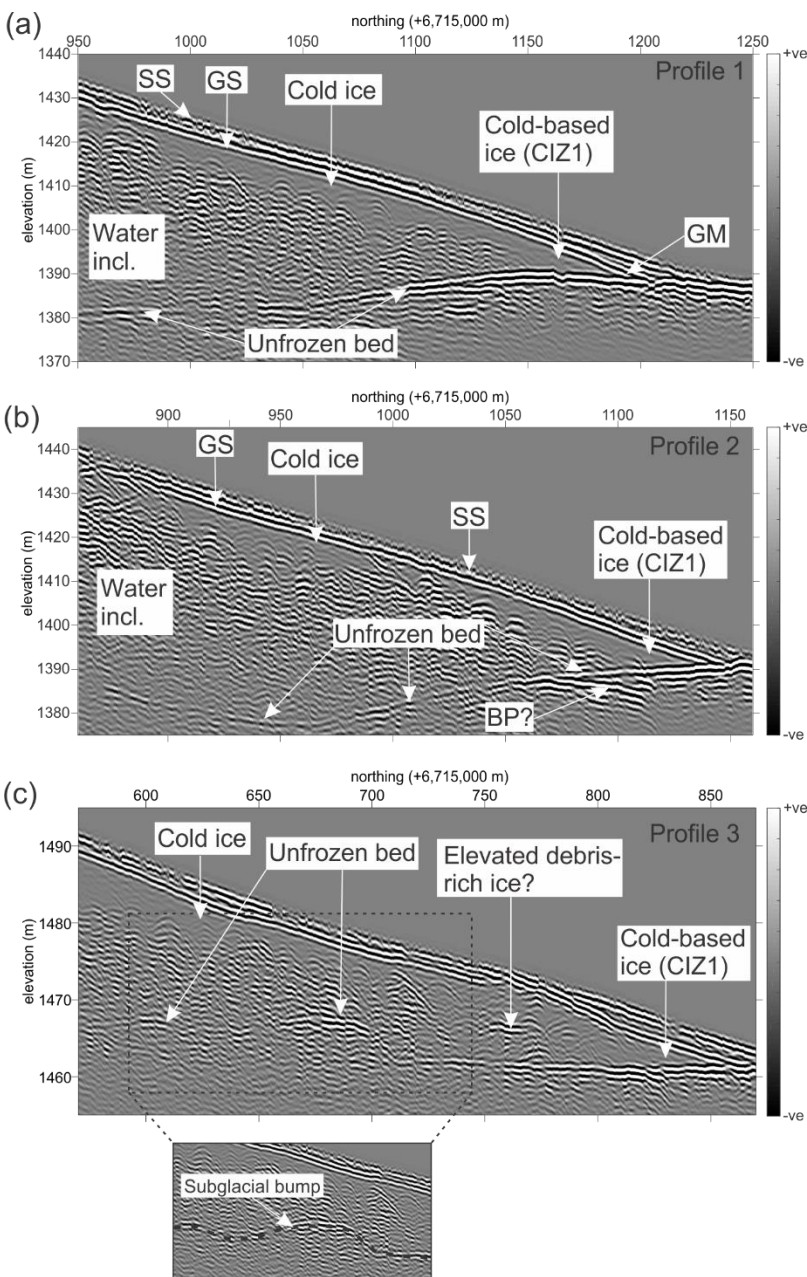

**Figure 5: GPR profiles from the Midtdalsbreen margin (locations in Fig. 4b, c), all parallel to ice flow. GS = glacier surface; GM = glacier margin; SS = snow surface. (a) PR1, acquired over the snout and glacier margin with upper cold-ice layer and cold-ice zone (CIZ) 1. Also labelled is the glacier bed and cold-based ice (CIZ1). Further up-flow, the glacier bed appears temperate, with evidence of water inclusions. (b) PR2, also acquired around the glacier snout. BP tentatively indicates the base of subglacial permafrost. (c) PR3, acquired around the south-eastern glacier snout. The glacier bed shows a possible topographic bump, with possible associated elevated debris-rich ice layers.**

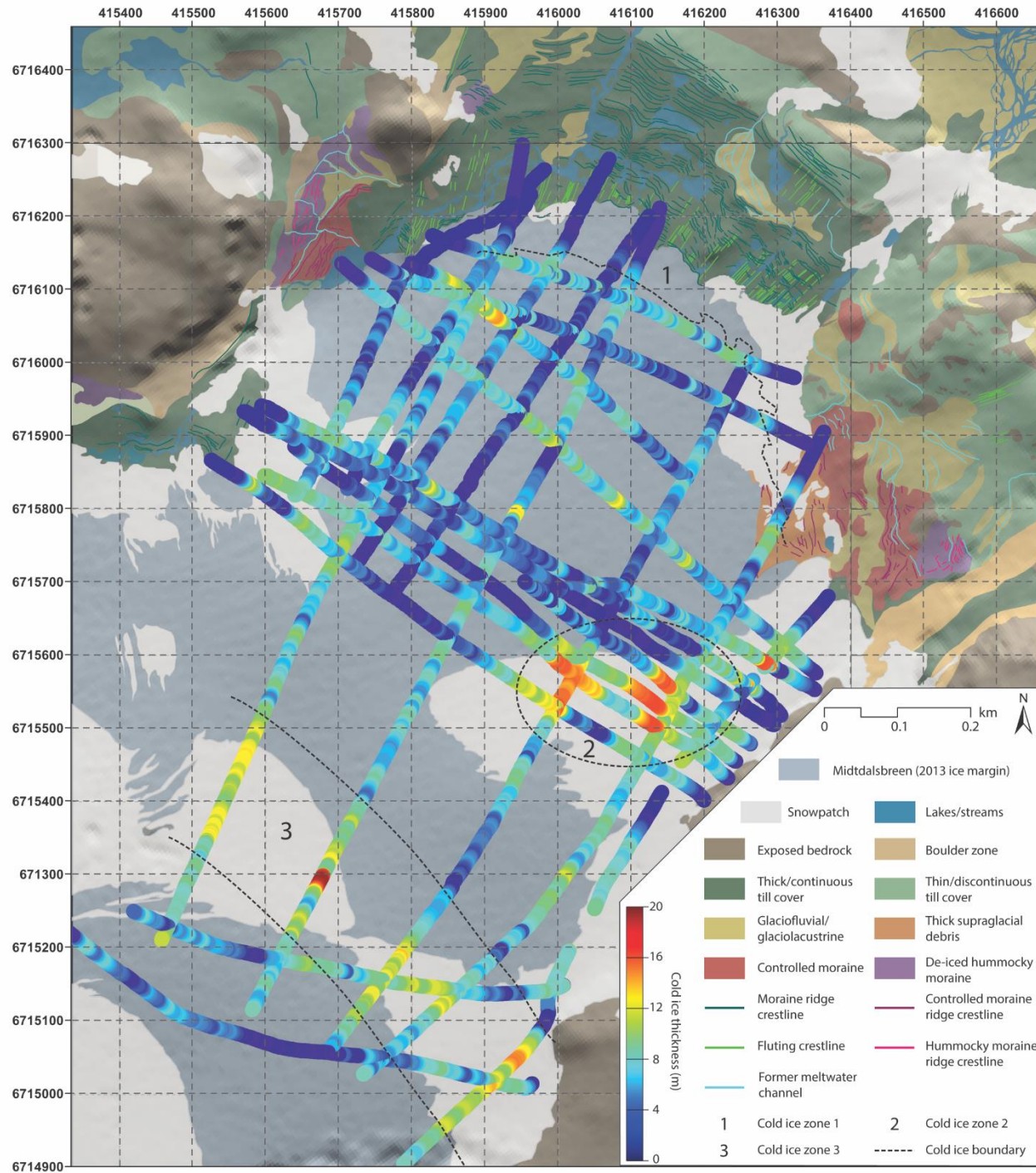

**Figure 6: Cold-ice thickness estimates on Midtdalsbreen from selected GPR profiles. Three cold-ice zones, CIZ1-CIZ3 are indicated. A corridor of 40 to 50 m of cold-based ice extending to the glacier bed is present at the glacier snout (CIZ1). Also included is a detailed geomorphological map of the glacier foreland.**

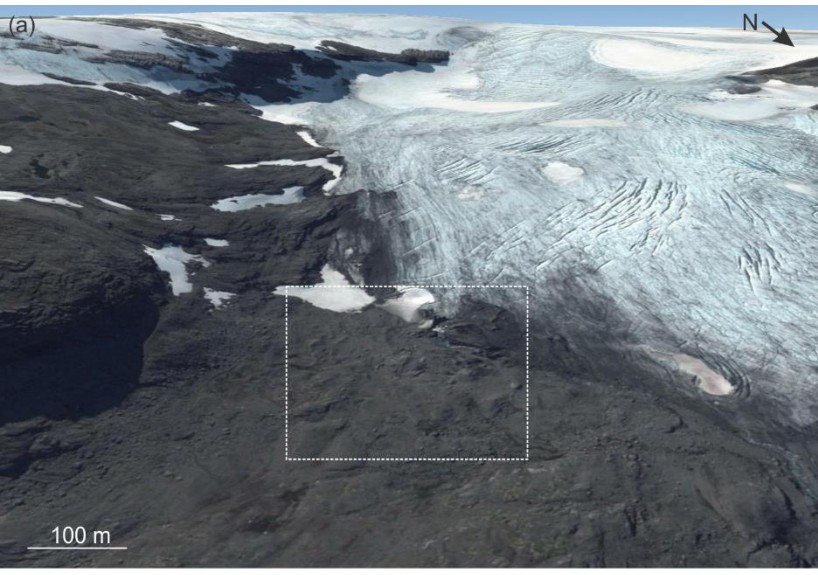

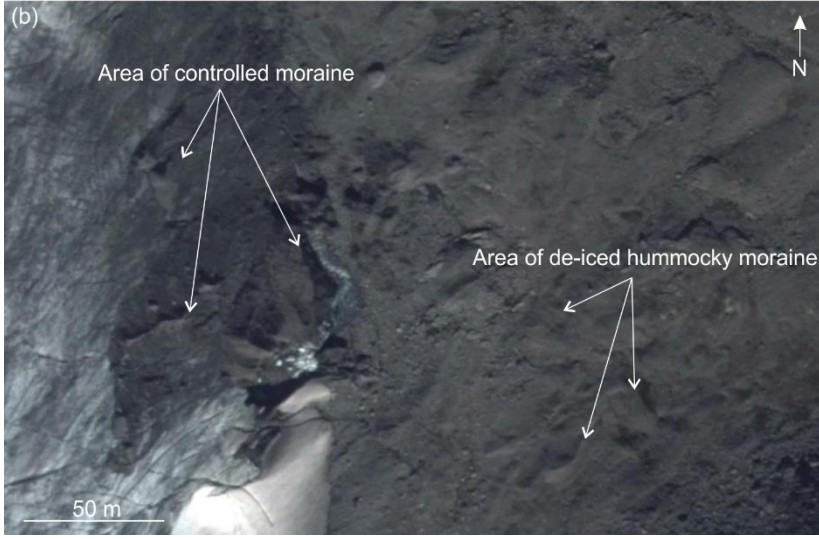

**Figure 7: (a) Google Earth satellite image (22/8/2013) over the south-eastern snout and foreland of Midtdalsbreen. Dashed box shows (b) area of supraglacial debris and controlled moraines (left) and de-iced hummocky moraine (right) discussed in text.**

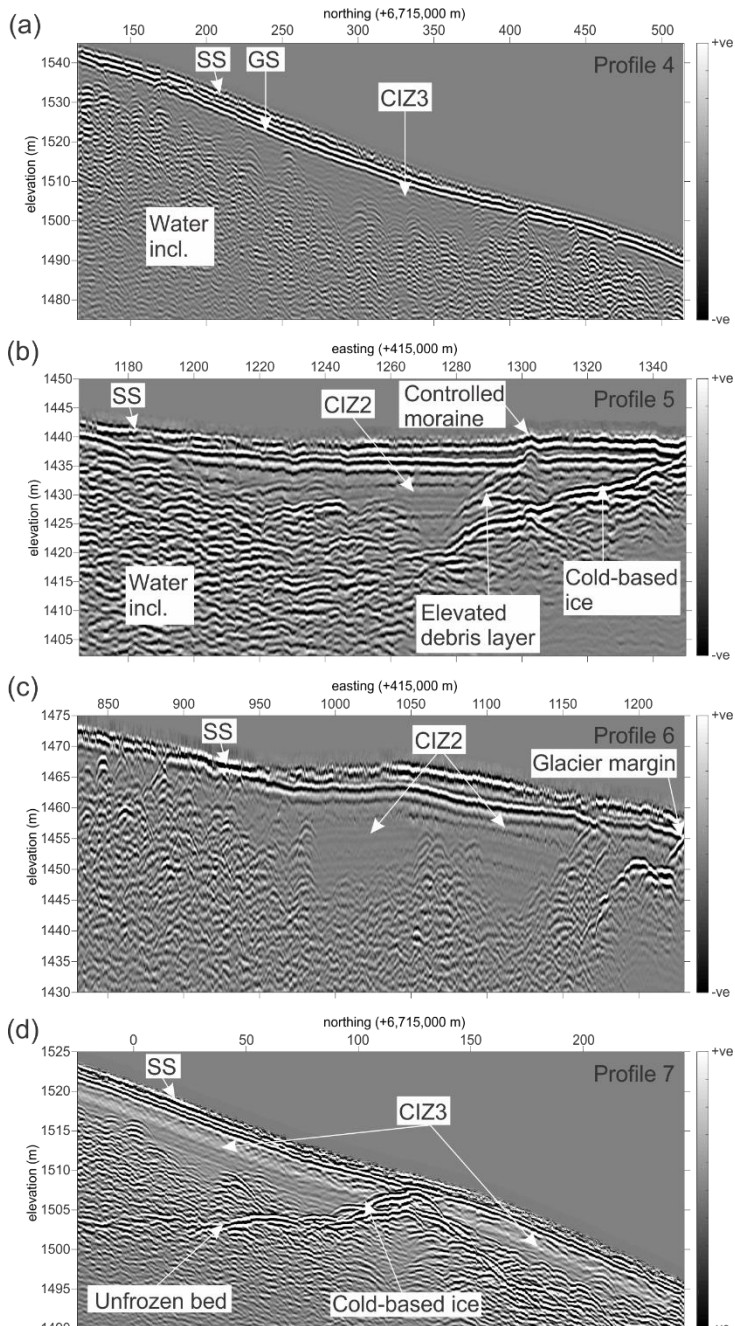

**Figure 8: GPR PR4-7 on Midtdalsbreen (Fig. 4b for locations). GS = glacier surface; SS = snow surface. (a) PR4, parallel to ice flow, showing cold-ice zone (CIZ) 3, ~15 m thick. (b) PR5, perpendicular to ice flow, indicating CIZ2 and a possible debris layer extending from close to the glacier bed to the ice surface. The collated mound could be a controlled moraine. The south east lateral margin of the glacier is located at the right of the profile. (c) PR6, perpendicular to ice flow, indicating CIZ2, ~15 m thick. (d) PR7, parallel to ice flow, indicating CIZ3, ~15 m thick. There is a well-defined cold-temperate transition surface (CTS) and a highly reflective glacier bed.**

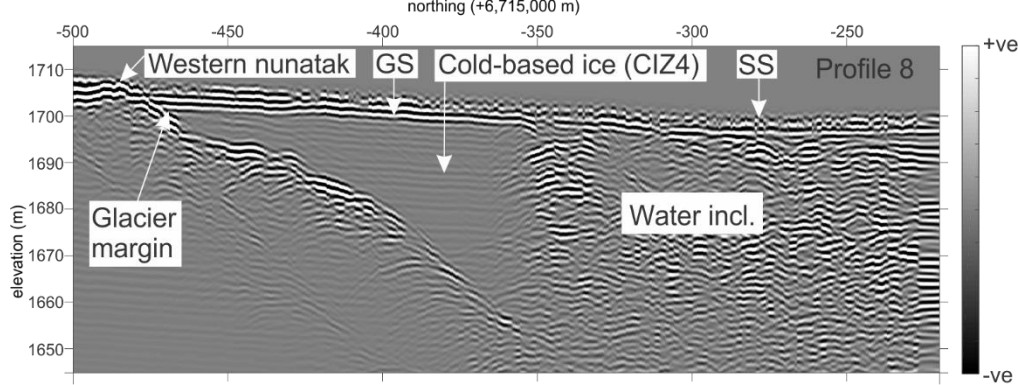

**Figure 9: PR8 (location in Fig 4a), close to the western nunatak. The transparent response of CIZ4 up to 30 m thick, extending to the glacier bed, can clearly be seen before the entire ice column becomes temperate. The strong dipping reflector is the glacier bed, not imaged below the temperate ice. GS = glacier surface; SS = snow surface.**

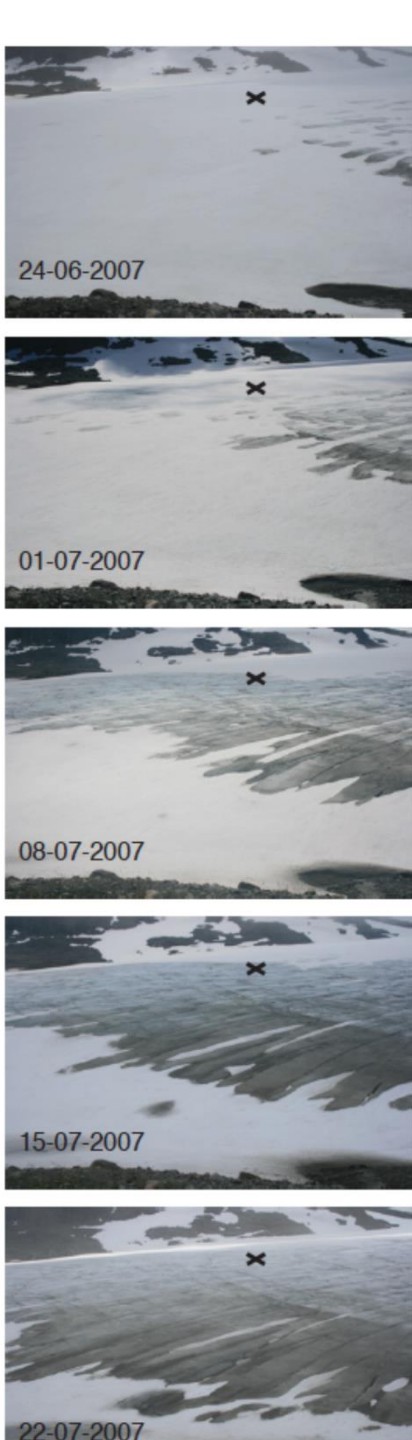

**Figure 10: Selection of photographs, taken by a camera set up by Giesen (2009) overlooking the Midtdalsbreen ablation area. The cross in the picture indicates an automatic weather station where Giesen (2009) measured ice velocity and ablation, as mentioned in the text.**

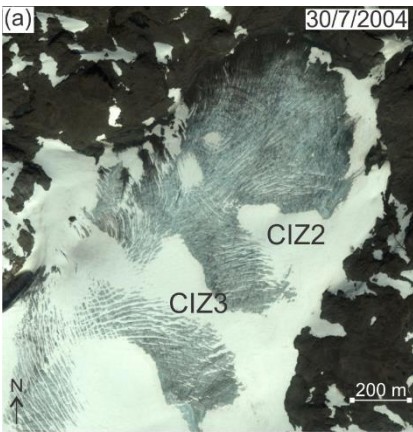

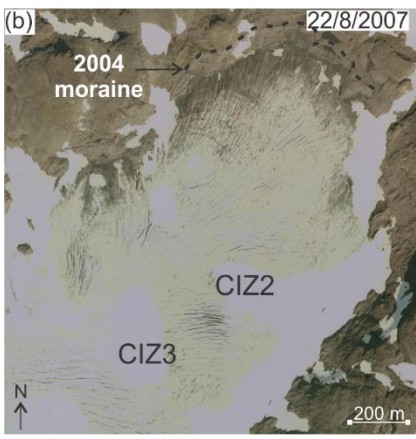

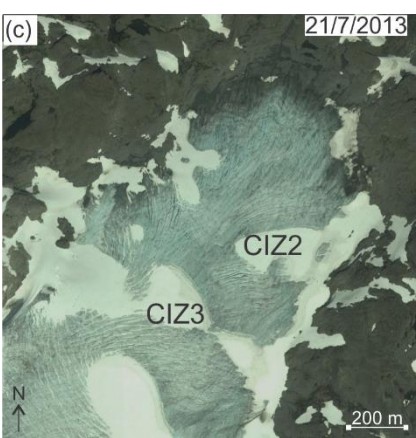

**Figure 11: Selection of satellite images during the late summer of 2004, 2007 and 2013 from the ablation area of Midtdalsbreen showing areas of late-lying seasonal snow patches that also correspond to CIZ2 and CIZ3. Sources (a) Google Earth, (b) Norge I Bilder, (c) Google Earth.**