# Peer review of "Pervasive cold-ice within a temperate glacier - implications for glacier thermal regime, sediment transport and foreland geomorphology"

_The Cryosphere, 2018_

## Referee Comment (RC1) · B. Etzelmuller (Referee) · 19 Nov 2018

The study by Reinardy et al investigates the thermal regime of an outlet glacier (Midtdalsbreen) emerging from the northern part of Hardangerjökull in southern Norway. The authors used a dense GPR network to map cold-ice patches on Midtdalsbreen. They identify cold zones at the glacier margin, often in connection with long-lasting snow patches, but also in the accumulation area in association to nunataks. The authors discuss these findings in relation to glacial geology and glacier dynamics. The paper builds on the observations of marginal moraine systems published in 2013 (Boreas)

[Figure]

where the main author relates the moraine architecture to cold-based marginal conditions. This manuscript focuses on the mapping of cold ice. The interpretations of the GPR survey are the main part of the study, and seems sound with reproducible results. In general, the paper is mostly well-written and certainly of interest for the community. However, there are several topics, which should be addressed: 1. Structure: The paragraph "Study area" should be divided from the introduction and should have an own number (2). The "Interpretation" chapter is a Discussion, and should be merged into the "Discussion" chapter. The "Conclusions" appear now more as a summary or a discussion part, I would suggest you give clear conclusions from your study. 2. General contents: I am not always happy with the use of references. The core of the study is the mapping of the cold ice patches in the marginal area of Midtdalsbreen, in which the authors claim it is the". . .first direct observations. . ." (see abstract). First of all, GPR surveying is not a direct observation. Second, this study is probably the first systematic mapping of cold ice on this glacier, however, direct measurements of glacier temperatures exists from earlier studies, unfortunately not published in international literature, but only as a thesis (in Norwegian) or "gray literature" in the form of excursion guides etc. E.g. Jon Ove Hagen has done his thesis in 1978 and measured directly the cold marginal area of Midtdalsbreen (Hagen, J. O. 1978. Brefrontprosesser ved Hardangerjökulen [Glacier front processes at Hardangerjøkulen]. Cand. real. Thesis, University of Oslo, Norway, Oslo), and attributed the marginal morphology to freezing processes etc. This information was later included in several publications, such as excursion guides (e.g. Liestøl, O. & Sollid, J. L. 1980. Glacier erosion and sedimentation at Hardangerjokulen and Omnsbreen. In: Orheim, O. (ed.), Symposium on Processes of Glacier Erosion and Sedimentation, Field Guide to Excursion. Norsk Polarinstitutt, Oslo, Geilo, Norway, 1-22.) or subsequent publications discussing the interaction between glaciers and permafrost in high mountain and arctic settings (e.g. Etzelmüller and Hagen 2005, Glacier-permafrost interaction in Arctic and alpine mountain environments with examples from southern Norway and Svalbard, Geological Society, London, Special Publications 2005; v. 242; p. 11-27, doi:10.1144/GSL.SP.2005.242.01.02, see

e.g. Fig. 1e). Most of these publications are cited in Reinardy et al 2013, and used as indications that the observed moraine pattern in the Boreas study was related to basal on-freezing (Fig. 9). I totally understand that literature in Norwegian is not necessarily known and certainly not understandable for international colleagues, however, some of the co-authors in this study might help. Concerning the important process of subglacial material entrainment, and also the possibility of material transport along shear planes, maybe you can also have a look on Weertman, J. 1961 (Mechanism for the formation of inner moraines found near the edge of cold ice caps and ice sheets. Journal of Glaciology, 3, 965 -978). 3. Glacier-permafrost interaction: The study in general generates very nice results and discussions, which are interesting in a glacier-permafrost interaction context. E.g., cold ice patches in connection to nunataks in the accumulation area is not surprising for the Finse area as these nunataks probably have permafrost (lower regional permafrost limit is c. 1400-1600 m a.s.l., depending on snow cover). Permafrost in mountain settings is of course a 3-dimensional problem (e.g. Nötzli et al 2007, JGR), and cold non-glaciated and maybe even snow-free areas influence adjacent glacier bodies thermally. If this is of interest, you may find relevant literature in e.g. Myhra, K. S., et al . ("Modelled Distribution and temporal Evolution of Permafrost in Steep Rock . . ..." PPP 28.1 (2017): 172-182). Another interesting topic is the influence of long-lasting snow patches as reason for cold ice development and persistence. I think the reasoning in the paper is fine. In the mountain-permafrost community, long-lasting and relatively stable snow patches have been used as permafrost indicators. Maybe some relevant literature is available within this topic also for the present study. There are also some recent activities about snow and permafrost in the Finse area you may find useful in your discussion. 4. Implications: The authors discuss implications of the cold-based areas for different topics. One is of course the sedimentology and moraine architecture. Here the authors rely much on the Boreas paper, which is fine, but avoid redundancies (also between introduction and discussion in this manuscript). Concerning the discussion on the influence on glacier dynamics, I wonder if a model sensitivity test could justify some of the proposed implications. Some of the co-authors

certainly have modelling experience from Hardangerjökul, and may help or indicate if such tests are difficult to perform. Another implication in the manuscript is related to the lake (Demmevatnet) dammed by Rembedalskåka. I wonder if this discussion is a bit out of scope of the paper and speculative. Probably you cannot compare thermal conditions at Rembedalskåka with Midtdalsbreen, where the first outlet glacier incl. Demmevatnet ends much lower than Midtdalsbreen (c. 1200 m a.s.l. or below, Midt-dalsbreen c. 1400+ m a.s.l.). At this elevation, permanent frozen conditions can only be expected in extremely shaded conditions, based on our experience on permafrost distribution both in steep snow-free rock walls and in more gentle, snow-covered terrains.

Some minor comments: P1, l 20: Consider avoiding term "... the first observation ....". P3, Study area: See comment above P6, l. 14: Delete this sentence, no need to know your plans for upcoming papers. P 8/9: Interpretation section – see comment above. Again, you have a 4.1. chapter without 4.2 etc, should be avoided and can included into a discussion.

---

## Referee Comment (RC2) · Waller (Referee) · 23 Nov 2018

General comment

This paper focuses on the presentation of extensive geophysical survey results that allow the authors to describe the thermal structure of this outlet glacier and to identify cold ice that is surprisingly extensive with the glacier being consider to be largely temperate in nature. The potential connection of this extensive cold ice with the progressive recession and downwasting of the glacier in response to climate change constitutes a

new and exciting finding that suggests that the thermal changes that are occurring in polythermal glaciers in high latitude locations may be more widespread that previously thought. In view of the wide ranging implications relating to glacier dynamic behavior and regional groundwater fluxes for example, this paper constitutes an important and novel contribution to the literature and as such I am strongly supportive of its publication within "The Cryosphere".

There are however some aspects of the paper that would benefit from further elaboration and clarification in order to clarify, emphasise and justify some of the key points reached. The paper as it stands for example seems somewhat "unbalanced" with a short results section being followed by more lengthy interpretation and discussion sections. The provision of more systematic detail within the results section coupled with greater synthesis and integration of the interpretation and discussion sections (as recommended by the other referee) would help in this regard. A series of more general comments are listed below before more specific comments are provided in relation to the specific sections of the paper.

1. The paper would benefit from a little more clarity in its description and discussion of thermal regime. Reference to Blatter & Hutter (1991) for example provides relevant discussion of the thermal structures of polythermal glaciers (that contain areas of cold ice). With this in mind, should Midtdalsbreen more accurately be defined as a polythermal glacier if it displays extensive areas of cold-based ice?

2. As noted by the other referee, this paper is potentially very relevant to ongoing discussions concerning the influential role played by glacier-permafrost interactions. It would be worth considering the thermal changes being discussed within this broader landscape context and identifying whether or not the glacier margin (or the nunataks) are associated with areas of permafrost that may extend beneath the margin to comprise areas of subglacial permafrost that relate to the areas of cold-based ice. I recommend that the authors consult the work of Etzelmüller & Hagan (2005) in particular (which also contains useful information thermal regimes). It is worth noting that the

controlled moraines described at the site have been explicitly connected to glacier-permafrost interactions, "…polythermal conditions are crucial to the concentration of supraglacial debris and controlled moraines in glacier snouts via processes that are most effective at the glacier-permafrost interface" (Evans, 2009, p183).

3. In considering the potential influence of cold-based ice on glacier dynamics, I would recommend consulting the work of Moore et al. (2011) on Storglaciaren. Whilst this glacier also featured cold-based marginal ice that previous authors had argued was responsible for longitudinal compression, thrusting and debris entrainment, Moore et al. provided convincing evidence that the cold-based ice had little dynamic influence in this situation (see comment below).

4. In considering the implications for sediment transport, I'd recommend the inclusion of more detail on the presence, nature and thickness of the basal ice present as the key product related to marginal cold-ice and basal adfreezing. Reference to basal ice review papers (e.g. Hubbard & Sharp, 1989 or Knight, 1997) as well as some observations in the results sections (see comment below) would help here.

Specific Comments

Abstract • Include specific reference to the methods used to determine the presence of cold ice – i.e. GPR surveys. • Suggest caution in the use of the term "frozen to the bed" (L22) as this risks conflating state (frozen) and temperature (cold). May be better to refer to the marginal ice being cold-based (i.e. the cold ice is interacting with the glacier bed).

1. Introduction • Explicitly identify the three types of glacier referred to in the opening sentence: E.g. warm-based (temperate), polythermal and cold-based (polar). • With this in mind, I would recommend differentiating between the thermal regime and the basal thermal regime. The focus within the opening paragraph is on the former whilst much of the focus has been on the later in view of its potential influence on subglacial hydrology, glacier dynamics and sediment transport. • P2, L21 – Does

GPR really constitute "direct measurement"? To me this would imply the measurement of temperature using thermistor strings for example whilst I would presume that GPR surveys are an indirect measurement relating to the state of water present in the ice. • P3, L5 – Presumably the basal freeze-on is responsible for the formation of debris-rich basal ice? If so it would be worth relating the process to this product that provides an important sediment transport pathway. • Section 1.1 - Is there any evidence of permafrost within the area? If so, it would be worth referring to within the second paragraph. • Consider including a figure that illustrates some of the key landform associations described in the text. This would be helpful with foreland geomorphology being one of the foci identified in the title.

2. Methods • Extensive detail is provided on the acquisition and processing of the GPR data but the section would benefit from a little more detail on the approaches used in terms of the geomorphological mapping and sedimentological analysis. E.g. o Mapping – Was this undertaking using remotely-sensed imagery, DEMs, field observations or a combination of the three? o Sedimentological analysis – What were the specific techniques employed and what were these applied to (e.g. debris within the ice and contained within specific landforms)?

3. Results • Suggest including a short paragraph that provides some descriptive detail on the basal ice present at the site (see general comment) – e.g. thickness and facies characteristics. This add relevant detail that connects the cold-based ice with a key process (basal adfreezing) that generates one of the glaciological implications (basal sediment transfer). • Figure 7 is clearly a key results figure but in having a dual purpose (to illustrate results relating to both the glacier margin and the foreland) it is somewhat compromised with the foreland geomorphology being rather hard to visualize. Suggest considering splitting this figure into two to enable the production of a larger figure that more clearly illustrates the spatial distribution of the glacial landsystems referred to in the text. • In addition, the provision of a little more systematic detail on the key landforms within the text would be welcome, particularly in relation

to the associations between the dead-ice topography, the hummocky moraine and the controlled moraines that are central to the focus of the paper.

3.1 Distribution of cold ice • L16-17 – Emphasise that this allows the delineation of areas of cold ice and warm ice (at the pmp). • This section would also benefit from a little more detail and in places a more systematic narrative. • P6, L24 – Just a comment – this is an interesting point in that it suggests that the survey was undertaken at a time when the cold ice is at its maximum extent raising the question of how much of this ice is a seasonal phenomenon and how much persists throughout the year. • P6, L26 – Clarify what is meant by "excess" in this context. • As mentioned earlier, I would caution against the use of the phrase "frozen to the bed" and would recommend the use of the phrase "cold-based ice". • P6, L29 – Suggest including an estimate of the width of the marginal zone of cold-based ice.

4. Interpretation • Opening sentence highlighting the use of the results to provide a detailed interpretation of glacier dynamics comes as a surprise with no velocity data having been provided. In view of the title and focus of the paper, isn't the key point here that the data can be used to describe and interpret the thermal structure of the glacier? • P7, L16-17 – If possible, it would be useful if this recent thinning could be quantified. • P7, L17 – Suggest adding a paragraph break to emphasise the switch in focus to the role played by seasonal snowcover. Within this section, clarify why late-lying snow patches would promote basal freeze-on. This could simply involve making the point sooner that the insulating influence of persistent snow patches will act to delay the warming of the ice surface during the summer. • P8, L1-2 – How confident can you be in the assertion that a sharp divide between chaotic and transparent areas of ice corresponds to a sharp thermal boundary? To be persuasive, I'd like to see more in the way of explanation here. If there haven't been any direct temperature measurements that have demonstrated this, then perhaps the inclusion of what you'd expect to see in the radargram if the boundary was diffuse would help highlight the positive evidence being used hear to reach the interpretation. • P8, L6-10 – It would be worth

relating this discussion of debris entrainment more explicitly to specific mechanisms of basal ice formation – for example in this case the process of basal adfreezing described initially by Weertman (1961) – see review by Hubbard & Sharp (1989) or Knight (1997).  c P8, L10-12 – Clarify why the water-saturated nature of the till precludes the formation of larger moraine ridges. Influence of debris flows? o Within this section, it would be worth citing some other literature that has advocated the same process in similar environments (e.g. Matthews et al., 1995).  c P8, L20-25 – An alternative hypothesis that has been proposed in the literature is that they related to the streaming of basal ice around subglacial obstacles (see Gordon et al., 1992).  c P8, para 2 – Are there observations of the debris bands or debris-rich folia within the marginal ice that Evans (2009) describes as being key to the formation of controlled moraines? In this respect, the observation of frozen-on sediment slabs described at the end of the paragraph is interesting – but would this lead to the formation of something rather different (i.e. it wouldn't necessarily generate the linear structures diagnostic of controlled moraines sensu stricto).  c P9, L6-8 – It is important to note here that the connection between marginal cold-based ice and glacier flow remains the subject of debate. Combined GPR surveys, temperature and velocity measurements at Storglaciaren for example Moore et al. (2011) led them conclude that cold ice is incapable of generating significant longitudinal stress and velocity gradients.  c Figure 10 – The controlled moraines referred to in the text and in the caption are not easily visible in figure 10c. Suggest including a smaller-scale (enlarged) version of this part of the glacier margin as part of this figure.

4.1 Cold-ice zones in the accumulation area  c The hypothesis that these cold ice zones are recent phenomena relating to downwasting seems rather conjectural in the absence of comparative historial data. The argument made here is entirely reasonable but if this is to be presented as result of the paper then more in the way of supporting evidence needs to be provided here (e.g. model predictions). Alternatively it could be identified as an area for further investigation.

5. Discussion • The suggestion that lateral drag might influence the glacier's dynamic behavior is interesting in relation to the aforementioned study by Moore et al. (2011). This paper argued that ice-bed coupling between cold-based marginal ice and unlithified sediments was incapable of exerting significant drag. However, a more rigid bed situation around the lateral margins might make this more likely. • The suggestion that ongoing climate change may be resulting in a change in the thermal regime of the glacier and an expansion in extent of cold ice and cold-based ice is for me the most important finding of the paper. There is a growing body of literature focusing on Svalbard (e.g. Lovell et al., 2015) that suggests that historical and ongoing climatic changes have resulted in a thinning and deceleration of glaciers that has in turn resulted in a reduction in the extent of warm-based ice and progressive shift from polythermal to entirely cold-based thermal regimes. This has wider implications for regional groundwater systems for example. This paper would however be the first to provide evidence that this counterintuitive change in glacier thermal regime (i.e. cooling in a warming climate) is also taking place in more temperate environments, with primarily temperate glaciers becoming polythermal.

6. Conclusion • I suggest that the conclusion focuses more on the observed thermal regime and the surprising extent of cold ice. Relating a potential change in thermal character akin to that which has been discussed in Svalbard (see previous bullet) would be a good way of emphasizing the significance of this contribution. • In relation to this, I think the authors need to be a little more cautious in their discussion of the implications for glacier dynamic behavior (L24-25). Whilst this is an entirely reasonable prediction, no velocity data or surface strain data is provided that demonstrate these changes. Perhaps this is best presented as a hypothesis and a focus for ongoing research at the site. • P11, L21 – Again, suggest referring simply to cold-based ice rather than ice being frozen to the bed.

Technical corrections • P6, L5 – If referring to the figures sequentially, shouldn't this this be figure 6? • P6, L17-20 – This sentence is complex and hard to follow.

[Figure]

Suggest splitting into two. • P9, L14-15 – Check syntax. Reword? • Figures: o 2c – Highlight the area depicted as a box in figure 1b. o 6 – Revise label in 6b to "controlled moraine". Also suggest re-labelling frozen bed as cold-based ice. o 7 – Caption – suggest using phrase cold-based ice. o 8 - Suggest re-labelling frozen bed as cold-based ice. o

Cited references

Blatter, H. & Hutter, K., 1991. Polythermal conditions in Arctic glaciers. Journal of Glaciology, 37(126), 261-269.

Etzelmüller, B. & Hagan, J., 2005. Glacier-permafrost interaction in Arctic and Alpine environments with examples from southern Norway and Svalbard. In: C. Harris & J.B. Murton (eds.), Cryospheric Systems: Glaciers & Permafrost, Geological Society of London Special Publication, 242, pp11-27.

Evans, D.J.A., 2009. Controlled moraines: origins, characteristics and palaeoglaciological implications. Quaternary Science Reviews, 28, 183-208.

Gordon, J.E. et al., 1992. The formation of glacial flutes: assessment of models with evidence from Lyngsdalen, North Norway. Quaternary Science Reviews, 11 (7-8), 709-731.

Hubbard, B. & Sharp, M.J., 1989. Basal ice formation and deformation: a review. Progress in Physical Geography, 529-558.

Knight, P.G., 1997. The basal ice layer of glaciers and ice sheets. Quaternary Science Reviews, 16, 975-993.

Lovell, H. et al., 2015. Former dynamic behavior of a cold-based valley glacier on Svalbard revealed by basal ice and structural glaciology investigations. Journal of Glaciology, 61 (226), 309-328.

Matthews, J.A. et al., 1995. Contemporary terminal-moraine ridge formation at a temperate glacier: Styggedalsbreen, Jotunheimen, southern Norway. Boreas, 24(2), 129-139.

Moore, P.L. et al., 2011. Effect of a cold margin on ice flow at the terminus of Storglaciaren, Sweden: implications for sediment transport. Journal of Glaciology, 57(201), 77-87.

---

## Short Comment (SC1) · 23 Nov 2018

The article deals with a very interesting issue of the interaction of so calle "cold-ice" and "temperate-ice". It presents very interesting and original research results and adds substantial detals to the understanding of glacier termcs. In the title, however, it also contains a reference to the transport of sediments and geomorphology of the forefield. Although the georadar does not show this directly, as B. Etzelmuller rightly pointed out in his review, the article describes two types of glacier thermics: related to pressure melting point (PMP) and below this temperature ie. frozen (to the bed) conditions.

[Figure]

I think that these thermal conditions visible on the glacier have their analogy also on its foreground. This means that the CTS is a surface that is essentially not limited to the glacier, but has its "thermal continuation" on the glacial forefield. This was also noted in similar sense by B. Etzelmuller indicating in point 3 of his review the existence of a glacier - permafrost relationship in which the role of CTS is also in some way visible. In this context, it is worth to analyzing Fig.1. from work: Etzelmüller, B., Hagen, J.O., 2005. Glacier-permafrost interaction in Arctic and alpine mountain with the examples of southern Norway and Svalbard. In: Harris, C., Murton, J.B. (Eds.), Cryospheric Systems: Glaciers and Permafrost Geological Society of London Special Publication No. 242. Geological Society of London, London, pp. 11-27. In fact the Authors combine in this paper base of permafrost (PB) with cold ice layer, and in fact indicate continuation of CTS and PB surface.

I allowed myself to develop this relationship based on geophysical research on Storglaciären, (where the CTS surface was first seen). The results of the study were published in the work: Dobiński et al. 2017. Cold-temperate transition surface and permafrost base (CTS-PB) as an environmental axis in glacier-permafrost relationship, based on research carried out on the Storglaciären and its forefield, northern Sweden, Quaternary Research, 88, 551-569. which may be interesting for authors.

best greetings

Wojciech Dobiński

---

## Author Comment (AC2) · 9 Jan 2019

**Response to reviewer comments (RC1)**

All reviews have been very helpful in significantly improving the manuscript and we agree with all the major changes that were suggested. The following changes were made in response to comments from Reviewer RC1 (B. Etzelműller) and are detailed in blue text. All page and line numbers refer to the updated revised manuscript while a second document is also provided with track changes marked.

The study by Reinardy et al investigates the thermal regime of an outlet glacier (Midtdalsbreen) emerging from the northern part of Hardangerjökull in southern Norway. The authors used a dense GPR network to map cold-ice patches on Midtdalsbreen. They identify cold zones at the glacier margin, often in connection with long-lasting snow patches, but also in the accumulation area in association to nunataks. The authors discuss these findings in relation to glacial geology and glacier dynamics. The paper builds on the observations of marginal moraine systems published in 2013 (Boreas) where the main author relates the moraine architecture to cold-based marginal conditions. This manuscript focuses on the mapping of cold ice. The interpretations of the GPR survey are the main part of the study, and seems sound with reproducible results. In general, the paper is mostly well-written and certainly of interest for the community. However, there are several topics, which should be addressed: 1. Structure: The paragraph "Study area" should be divided from the introduction and should have an own number (2). The "Interpretation" chapter is a Discussion, and should be merged into the "Discussion" chapter. The "Conclusions" appear now more as a summary or a discussion part, I would suggest you give clear conclusions from your study.

We have followed all the above recommendations regarding the structure of the paper and reordered the sections accordingly. In addition we have totally rewritten the conclusion (P14 L18-P15 L2)

2. General contents: I am not always happy with the use of references. The core of the study is the mapping of the cold ice patches in the marginal area of Midtdalsbreen, in which the authors claim it is the". . .first direct observations. . ." (see abstract). First of all, GPR surveying is not a direct observation. Second, this study is probably the first systematic mapping of cold ice on this glacier, however, direct measurements of glacier temperatures exists from earlier studies, unfortunately not published in international literature, but only as a thesis (in Norwegian) or "gray literature" in the form of excursion guides etc. E.g. Jon Ove Hagen has done his thesis in 1978 and measured directly the cold marginal area of Midtdalsbreen (Hagen, J. O. 1978. Brefrontprosesser ved Hardangerjökulen [Glacier front processes at Hardangerjøkulen]. Cand. real. Thesis, University of Oslo, Norway, Oslo), and attributed the marginal morphology to freezing processes etc. This information was later included in several publications, such as excursion guides (e.g. Liestøl, O. & Sollid, J. L. 1980. Glacier erosion and sedimentation at Hardangerjokulen and Omnsbreen. In: Orheim, O. (ed.), Symposium on Processes of Glacier Erosion and Sedimentation, Field Guide to Excursion. Norsk Polarinstitutt, Oslo, Geilo, Norway, 1-22.) or subsequent publications discussing the interaction between glaciers and permafrost in high mountain and arctic settings (e.g. Etzelmüller and Hagen 2005, Glacier-permafrost interaction in Arctic and alpine mountain environments with examples from southern Norway and Svalbard, Geological Society, London, Special Publications 2005; v. 242; p. 11-27, doi:10.1144/GSL.SP.2005.242.01.02, see e.g. Fig. 1e). Most of these publications are cited in Reinardy et al 2013, and used as indications that the observed moraine pattern in the Boreas study was related to basal on-freezing (Fig. 9). I totally understand that literature in Norwegian is not necessarily known and certainly not understandable for international colleagues, however, some of the co-authors in this study might help. Concerning the

important process of subglacial material entrainment, and also the possibility of material transport along shear planes, maybe you can also have a look on Weertman, J. 1961 (Mechanism for the formation of inner moraines found near the edge of cold ice caps and ice sheets. Journal of Glaciology, 3, 965 -978).

We follow the above recommendations and have now incorporated all suggested literature highlighting the results of earlier studies into the thermal regime and permafrost distribution at Midtdalsbreen (P4 L3-7; P9 L9-10; P10 L6-8, P10 L31-32). We also now indicate what we believe to be the base permafrost in GPR profile 2 in our new Figure 5b which fits the thermal regime model suggested by Etzelműller and Hagen (2005, their Fig. 1e) (P10 L6-11). We also now include discussion of debris entrainment (P9 L31-P10 L14) with reference to Weertman (1961) and sediment elevation (P11 L4-25).

3. Glacier-permafrost interaction: The study in general generates very nice results and discussions, which are interesting in a glacier-permafrost interaction context. E.g., cold ice patches in connection to nunataks in the accumulation area is not surprising for the Finse area as these nunataks probably have permafrost (lower regional permafrost limit is c. 1400-1600 m a.s.l., depending on snow cover). Permafrost in mountain settings is of course a 3-dimensional problem (e.g. Nötzli et al 2007, JGR), and cold non-glaciated and maybe even snow-free areas influence adjacent glacier bodies thermally. If this is of interest, you may find relevant literature in e.g. Myhra, K. S., et al . ("Modelled Distribution and temporal Evolution of Permafrost in Steep Rock . . ..." PPP 28.1 (2017): 172-182). Another interesting topic is the influence of long-lasting snow patches as reason for cold ice development and persistence. I think the reasoning in the paper is fine. In the mountain-permafrost community, longlasting and relatively stable snow patches have been used as permafrost indicators. Maybe some relevant literature is available within this topic also for the present study. There are also some recent activities about snow and permafrost in the Finse area you may find useful in your discussion.

We have followed all the above recommendations and have now included far more discussion on the distribution of permafrost (including the suggested references), the potential influence of permafrost on glacier thermal regime (P11 L31-P12 L11) and the interaction of the cold-temperate surface (CTS) with the base of the permafrost at the glacier snout (P10 L6-12). We also include literature discussing the influence of variable snow cover/depth on ground thermal regime (P9 L8-15) as well as basal motion.

4. Implications: The authors discuss implications of the cold-based areas for different topics. One is of course the sedimentology and moraine architecture. Here the authors rely much on the Boreas paper, which is fine, but avoid redundancies (also between introduction and discussion in this manuscript). Concerning the discussion on the influence on glacier dynamics, I wonder if a model sensitivity test could justify some of the proposed implications. Some of the co-authors certainly have modelling experience from Hardangerjökul, and may help or indicate if such tests are difficult to perform. Another implication in the manuscript is related to the lake (Demmevatnet) dammed by Rembedalskåka. I wonder if this discussion is a bit out of scope of the paper and speculative. Probably you cannot compare thermal conditions at Rembedalskåka with Midtdalsbreen, where the first outlet glacier incl. Demmevatnet ends much lower than Midtdalsbreen (c. 1200 m a.s.l. or below, Midtdalsbreen c. 1400+ m a.s.l.). At this elevation, permanent frozen conditions can only be expected in extremely shaded conditions, based on our experience on permafrost distribution both in steep snow-free rock walls and in more gentle, snow-covered terrains.

We try now to focus more on the controlled moraines and other landforms within the south-eastern glacier foreland (P6 L32-P7 L33) that have not been previously described in Reinardy et al. (2013). We

have also removed or toned down references to glacier dynamics including changing the title of the paper (P1 L1-2). We have removed the discussion relating to Nedre Demmevatnet.

Some minor comments: P1, l 20: Consider avoiding term ". . . the first observation . . ..". P3, Study area: See comment above P6, l. 14: Delete this sentence, no need to know your plans for upcoming papers. P 8/9: Interpretation section – see comment above. Again, you have a 4.1. chapter without 4.2 etc, should be avoided and can included into a discussion.

All suggested minor comments have been applied.

---

## Author Comment (AC3) · 9 Jan 2019

**Response to reviewer comments (RC2)**

All reviews have been very helpful in significantly improving the manuscript and we agree with all the major changes that were suggested. The following changes were made in response to comments from Reviewer RC2 (R. Waller) and are detailed in blue text. All page and line numbers refer to the updated revised manuscript while a second document is also provided with track changes marked.

General comment

This paper focuses on the presentation of extensive geophysical survey results that allow the authors to describe the thermal structure of this outlet glacier and to identify cold ice that is surprisingly extensive with the glacier being consider to be largely temperate in nature. The potential connection of this extensive cold ice with the progressive recession and downwasting of the glacier in response to climate change constitutes a new and exciting finding that suggests that the thermal changes that are occurring in polythermal glaciers in high latitude locations may be more widespread that previously thought. In view of the wide ranging implications relating to glacier dynamic behaviour and regional groundwater fluxes for example, this paper constitutes an important and novel contribution to the literature and as such I am strongly supportive of its publication within "The Cryosphere".

There are however some aspects of the paper that would benefit from further elaboration and clarification in order to clarify, emphasise and justify some of the key points reached. The paper as it stands for example seems somewhat "unbalanced" with a short results section being followed by more lengthy interpretation and discussion sections. The provision of more systematic detail within the results section coupled with greater synthesis and integration of the interpretation and discussion sections (as recommended by the other referee) would help in this regard.

The results section now outlines more detail as requested (P6 L26-P7 L33), see detailed responses below.

A series of more general comments are listed below before more specific comments are provided in relation to the specific sections of the paper.

1. The paper would benefit from a little more clarity in its description and discussion of thermal regime. Reference to Blatter & Hutter (1991) for example provides relevant discussion of the thermal structures of polythermal glaciers (that contain areas of cold ice). With this in mind, should Midtdalsbreen more accurately be defined as a polythermal glacier if it displays extensive areas of cold-based ice?

Glacier thermal regime and basal thermal regime are now defined with more detail and reference to Blatter and Hutter (1991) (P2 L2-15). We now make the point in the Abstract (P1), Discussion (P8) and Conclusion (14) sections that indeed Midtdalsbreen may become polythermal.

2. As noted by the other referee, this paper is potentially very relevant to ongoing discussions concerning the influential role played by glacier-permafrost interactions. It would be worth considering the thermal changes being discussed within this broader landscape context and identifying whether or not the glacier margin (or the nunataks) are associated with areas of permafrost that may extend beneath the margin to comprise areas of subglacial permafrost that relate to the areas of cold-based ice. I recommend that the authors consult the work of Etzelmüller & Hagan (2005)

in particular (which also contains useful information thermal regimes). It is worth noting that the controlled moraines described at the site have been explicitly connected to glacierpermafrost interactions, ". . .polythermal conditions are crucial to the concentration of supraglacial debris and controlled moraines in glacier snouts via processes that are most effective at the glacier-permafrost interface" (Evans, 2009, p183).

The importance of permafrost in the study area is now highlighted. Several relevant references relating to glacier-permafrost interaction have now been added. Specifically, we have now included a discussion on possible sediment accretion and elevation at the cold temperate interface (CTS) (P9 L31-P10 L13; P11 L4-25) and the relationship of the CTS with the base of the permafrost underneath the glacier snout which was previously modelled by Etzelmüller and Hagen, (2005 now also referenced in the main body of text P10 L5-L12). We have also included some discussion of the basal ice layer which was previously mentioned in Reinardy et al. (2013) (P7 L8-17) and highlighted the importance of the glacier-permafrost interaction in the deposition of the supraglacial debris cover and the controlled moraines (P7 L13-15). This added discussion of glacier-permafrost interaction also led us to have another look at our GPR data from the glacier snout area. In GPR profile 2, Fig. 5b we observed dipping reflectors below the ice-bed interface which we now interpret as the base of the permafrost thus supporting the model of Etzelmüller and Hagen (2005) (P10 L3-11).

3. In considering the potential influence of cold-based ice on glacier dynamics, I would recommend consulting the work of Moore et al. (2011) on Storglaciaren. Whilst this glacier also featured cold-based marginal ice that previous authors had argued was responsible for longitudinal compression, thrusting and debris entrainment, Moore et al. provided convincing evidence that the cold-based ice had little dynamic influence in this situation (see comment below).

We were not aware of the paper by Moore et al. (2011) but it does indeed have some very interesting implications for our work. Specifically, Moore et al. (2011) highlight the importance of subglacial topography in the elevation of subglacial debris to englacial and supraglacial positions. We checked the extensive GPR data we have from the front of Midtdalsbreen and, similar to Moore et al. we also found a subglacial "bump" in GPR profile 3 which we now show in the new Fig. 5c. We now discuss the role of this bump in terms of debris elevation (P11 L13-25).

4. In considering the implications for sediment transport, I'd recommend the inclusion of more detail on the presence, nature and thickness of the basal ice present as the key product related to marginal cold-ice and basal adfreezing. Reference to basal ice review papers (e.g. Hubbard & Sharp, 1989 or Knight, 1997) as well as some observations in the results sections (see comment below) would help here.

Observations of the basal ice layer are primarily based on those of Reinardy et al. (2013) from the south-eastern margin of Midtdalsbreen. We have now also added some descriptions of the basal ice layer in the Results section (P7 L8-L13) and discuss the basal ice layer in terms of debris entrainment including reference to Knight (1997) (P9 L34-P10 L3).

Specific Comments

Abstract â˘A ´c Include specific reference to the methods used to determine the presence of cold ice – i.e. GPR surveys. â˘A´c Suggest caution in the use of the term "frozen to the bed" (L22) as this risks conflating state (frozen) and temperature (cold). May be better to refer to the marginal ice being cold-based (i.e. the cold ice is interacting with the glacier bed).

Reference to mapping of cold-ice using GPR is made and the term "cold-based ice" is now used as suggested (P1).

1. Introduction â˘A ´c Explicitly identify the three types of glacier referred to in the opening sentence: E.g. warm-based (temperate), polythermal and cold-based (polar). â˘A ´c With this in mind, I would recommend differentiating between the thermal regime and the basal thermal regime. The focus within the opening paragraph is on the former whilst much of the focus has been on the later in view of its potential influence on subglacial hydrology, glacier dynamics and sediment transport. â˘A´c P2, L21 – Does GPR really constitute "direct measurement"? To me this would imply the measurement of temperature using thermistor strings for example whilst I would presume that GPR surveys are an indirect measurement relating to the state of water present in the ice. â˘A ´c P3, L5 – Presumably the basal freeze-on is responsible for the formation of debris rich basal ice? If so it would be worth relating the process to this product that provides an important sediment transport pathway. â˘A ´c Section 1.1 - Is there any evidence of permafrost within the area? If so, it would be worth referring to within the second paragraph. â˘A ´c Consider including a figure that illustrates some of the key landform associations described in the text. This would be helpful with foreland geomorphology being one of the foci identified in the title.

Definitions of thermal regimes have now been added, the term "direct measurement" has been removed, and discussion of basal freeze-on supplying sediment to the basal ice layer is now included (P9 L34-P10 L3). The presence of permafrost in the area is now explicitly identified and discussed (P4 L3-7; P7 L13-L19; P10 L3-11; P10 L30-P11 L2; P11 L31-P12 L6). We have now also included an extra figure (Fig. 7) showing the key landforms. We also discuss the continuum of controlled moraines to de-iced hummocky moraine shown in the geomorphological map (Fig. 6) and Figure 7 (P6 L33-P7 L2; P14 L26-28).

2. Methods â˘A ´c Extensive detail is provided on the acquisition and processing of the GPR data but the section would benefit from a little more detail on the approaches used in terms of the geomorphological mapping and sedimentological analysis. E.g. o Mapping – Was this undertaking using remotely-sensed imagery, DEMs, field observations or a combination of the three? O Sedimentological analysis – What were the specific techniques employed and what were these applied to (e.g. debris within the ice and contained within specific landforms)?

More detail regarding the sedimentological and geomorphological work carried out is now included as suggested (P6 L15-P7 L33).

3. Results â˘A ´c Suggest including a short paragraph that provides some descriptive detail on the basal ice present at the site (see general comment) – e.g. thickness and facies characteristics. This add relevant detail that connects the cold-based ice with a key process (basal adfreezing) that generates one of the glaciological implications (basal sediment transfer). â˘A ´c Figure 7 is clearly a key results figure but in having a dual purpose (to illustrate results relating to both the glacier margin and the foreland) it is somewhat compromised with the foreland geomorphology being rather hard to visualize. Suggest considering splitting this figure into two to enable the production of a larger figure that more clearly illustrates the spatial distribution of the glacial landsystems referred to in the text. â˘A ´c In addition, the provision of a little more systematic detail on the key landforms within the text would be welcome, particularly in relation to the associations between the dead-ice topography, the hummocky moraine and the controlled moraines that are central to the focus of the paper.

Exposure of the basal ice layer was limited but is briefly described partly from previous work carried out by Reinardy et al. (2013) (P7 L8-10). We have now included an new figure (Fig. 7) that clearly highlights the area of controlled moraines linked to debris-rich glacial septa visible in some cases and de-iced hummocky moraines on the foreland. This extra figure (P31) now clearly illustrates the geomorphology and the spatial distribution of the different landforms on the south-eastern glacier snout, margin and foreland. The association of the controlled moraine and production of de-iced hummocky moraine through ice-core melt out is now also clearly highlighted (P6 L33-P7 L2; P14 L26-28).

.

3.1 Distribution of cold ice ăˇA ´c L16-17 – Emphasise that this allows the delineation of areas of cold ice and warm ice (at the pmp). ăˇA´c This section would also benefit from a little more detail and in places a more systematic narrative. ăˇA ´c P6, L24 – Just a comment – this is an interesting point in that it suggests that the survey was undertaken at a time when the cold ice is at its maximum extent raising the question of how much of this ice is a seasonal phenomenon and how much persists throughout the year. ăˇA ´c P6, L26 – Clarify what is meant by "excess" in this context. ăˇA ´c As mentioned earlier, I would caution against the use of the phrase "frozen to the bed" and would recommend the use of the phrase "cold-based ice". ăˇA´c P6, L29 – Suggest including an estimate of the width of the marginal zone of cold-based ice.

We have now separated the Results into section 4.1 "Geomorphology and sedimentology of the foreland" (P6 L27-P7 L33) and section 4.2 "Distribution of cold ice" (P8 L1-27). We have also split the Discussion section into four subsections (5.1-5.4) and added a significant amount of new detail (P8 L28-P14 L16). We adjusted the terminology as suggested.

4. Interpretation ăˇA ´c Opening sentence highlighting the use of the results to provide a detailed interpretation of glacier dynamics comes as a surprise with no velocity data having been provided. In view of the title and focus of the paper, isn't the key point here that the data can be used to describe and interpret the thermal structure of the glacier? ăˇA´c P7, L16-17 – If possible, it would be useful if this recent thinning could be quantified. ăˇA ´c P7, L17 – Suggest adding a paragraph break to emphasise the switch in focus to the role played by seasonal snowcover. Within this section, clarify why latelying snow patches would promote basal freeze-on. This could simply involve making the point sooner that the insulating influence of persistent snow patches will act to delay the warming of the ice surface during the summer. ăˇA ´c P8, L1-2 – How confident can you be in the assertion that a sharp divide between chaotic and transparent areas of ice corresponds to a sharp thermal boundary? To be persuasive, I'd like to see more in the way of explanation here. If there haven't been any direct temperature measurements that have demonstrated this, then perhaps the inclusion of what you'd expect to see in the radargram if the boundary was diffuse would help highlight the positive evidence being used hear to reach the interpretation. ăˇA ´c P8, L6-10 – It would be worth relating this discussion of debris entrainment more explicitly to specific mechanisms of basal ice formation – for example in this case the process of basal adfreezing described initially by Weertman (1961) – see review by Hubbard & Sharp (1989) or Knight (1997). ăˇA ´c P8, L10-12 – Clarify why the water-saturated nature of the till precludes the formation of larger moraine ridges. Influence of debris flows? o Within this section, it would be worth citing some other literature that has advocated the same process in similar environments (e.g. Matthews et al., 1995). ăˇA ´c P8, L20-25 – An alternative hypothesis that has been proposed in the literature is that they related to the streaming of basal ice around subglacial obstacles (see Gordon et al., 1992). ăˇA ´c P8, para 2 – Are there observations of the debris bands or debris-rich folia within the marginal ice that Evans (2009) describes as being key to the formation of controlled moraines? In this respect, the observation of frozen-on sediment slabs described at the end of the paragraph is interesting – but would this lead to the formation of something rather different (i.e. it wouldn't necessarily generate the linear structures diagnostic of controlled moraines sensu stricto). ăˇA ´c P9, L6-8 – It is important to note here that the connection between marginal cold-based ice and glacier flow remains the subject of debate. Combined GPR surveys, temperature and velocity measurements at Storglaciaren for example Moore et al. (2011) led them conclude that cold ice is incapable of generating significant longitudinal stress and velocity gradients. ăˇA ´c Figure 10 – The controlled moraines referred to in the text and in the caption are not easily visible in figure 10c. Suggest including a smaller-scale (enlarged) version of this part of the glacier margin as part of this figure.

We have added more detail to the Results section (P6 L26-P8 L27) and parts of our interpretations have been integrated into the Discussion section (P8 L28-P14 L16) as requested by Reviewer 1.

Reference to glacier dynamics has now been removed from the title (P1 L1-2) and toned down elsewhere as no long term glacier mass balance records are available for Midtdalsbreen although thinning rates were measured between 1961 to 1995 and have now been included (P12 L6-9). We have now also expanded the discussion on snow cover and the relationship with ground/ice surface temperatures (P4 L3-7; P7 L13-19; P9 L8-15; P10 L30-P11 L2; P11 L31-P12 L6). The discussion about the CTS has been modified to clearly link the CTS with underlying permafrost (P10 L3-11). We have included discussion of debris entrainment (P9 L31-P10 L3) and elevation (P11 L4-25) considering the CTS (P10 L3-11), permafrost and subglacial topography (P10 L13-25) as well as degradation of moraines (P7 L26-31; P10 L32-33; P14 L26-28). We include reference to Gordon et al. 1992 regarding streaming ice flow and formation of flutes (P10 L18-20). We discuss fully the results of Moore et al. (2011) in relation to our findings at Midtdalsbreen which now include a description of a subglacial "bump" displayed in the new Figure 5c (P11 L13-25). We include a new Figure 7 which clearly shows the moraines discussed in the text (P31).

4.1 Cold-ice zones in the accumulation area ˘A´c The hypothesis that these cold ice zones are recent phenomena relating to downwasting seems rather conjectural in the absence of comparative historial data. The argument made here is entirely reasonable but if this is to be presented as result of the paper then more in the way of supporting evidence needs to be provided here (e.g. model predictions). Alternatively it could be identified as an area for further investigation.

We have toned down our assertions regarding glacier downwasting but include some thinning rates measured between 1961-1995 and highlight that the nunataks are above the altitudinal permafrost limit and thus likely have a significant influence on the formation of cold-ice in CIZ4-5 (P11 L31-P12 L11).

5. Discussion ˘A ´c The suggestion that lateral drag might influence the glacier's dynamic behavior is interesting in relation to the aforementioned study by Moore et al. (2011). This paper argued that ice-bed coupling between cold-based marginal ice and unlithified sediments was incapable of exerting significant drag. However, a more rigid bed situation around the lateral margins might make this more likely. ˘A ´c The suggestion that ongoing climate change may be resulting in a change in the thermal regime of the glacier and an expansion in extent of cold ice and cold-based ice is for me the most important finding of the paper. There is a growing body of literature focusing on Svalbard (e.g. Lovell et al., 2015) that suggests that historical and ongoing climatic changes have resulted in a thinning and deceleration of glaciers that has in turn resulted in a reduction in the extent of warm-based ice and progressive shift from polythermal to entirely cold-based thermal regimes. This has wider implications for regional groundwater systems for example. This paper would however be the first to provide evidence that this counterintuitive change in glacier thermal regime (i.e. cooling in a warming climate) is also taking place in more temperate environments, with primarily temperate glaciers becoming polythermal.

We have now included a discussion of our work in relation to that of Moore et al. (2011) (P11 L13-25). We also now state the possibility of expansion of cold-ice in a warming climate in the Abstract (P1 L27-30), Discussion (P12 L16-18; P14 L3-10) and Conclusion (P14 L33-P15 L2) sections of the paper highlighting this as a key result of the study.

6. Conclusion ˘A ´c I suggest that the conclusion focuses more on the observed thermal regime and the surprising extent of cold ice. Relating a potential change in thermal character akin to that which has been discussed in Svalbard (see previous bullet) would be a good way of emphasizing the significance of this contribution. ˘A ´c In relation to this, I think the authors need to be a little more cautious in their discussion of the implications for glacier dynamic behavior (L24-25). Whilst this is an entirely reasonable prediction, no velocity data or surface strain data is provided that demonstrate these changes. Perhaps this is best presented as a hypothesis and a focus for ongoing research at the

site. ăˇA ´c P11, L21 – Again, suggest referring simply to cold-based ice rather than ice being frozen to the bed.

The conclusion has been entirely rewritten following the above suggestions and those of Reviewer 1 (P14 L19-P15 L2). Specifically we have removed text relating to glacier dynamics and focus on the key results from both the GPR and geological data presented.

Technical corrections ăˇA´c P6, L5 – If referring to the figures sequentially, shouldn't this this be figure 6? ăˇA ´c P6, L17-20 – This sentence is complex and hard to follow.

Suggest splitting into two. ăˇA ´c P9, L14-15 – Check syntax. Reword? ăˇA ´c Figures: o 2c – Highlight the area depicted as a box in figure 1b. o 6 – Revise label in 6b to "controlled moraine". Also suggest re-labelling frozen bed as cold-based ice. o 7 – Caption – suggest using phrase cold-based ice. o 8 - Suggest re-labelling frozen bed as cold-based ice. o

All suggested technical corrections have been applied.

---

## Author Comment (AC4) · 9 Jan 2019

Our updated revised manuscript now includes far more discussion on the glacier-permafrost interaction and the connection between the cold-temperate surface (CTS) and the base permafrost is highlighted (P9 L31-P10 L13). We also identified what we believe to be the base permafrost in GPR profile 2 (new Fig. 5b) that links with the CTS and the location of the freezing isotherm. The recommended paper of Dobiński et al., (2017) was particularly useful in our discussion of the CTS which we have now also included in our references in addition to EtzelmÅśller and Hagen (2005).

---

## Author Response (AR2)

**Pervasive cold-ice within a temperate glacier - implications for glacier thermal regime, sediment transport and foreland geomorphology**

5  **Summary of changes:** We thank the editor for the approval of our manuscript. As required, we have uploaded our archive of GPR data to a FIAR-compliant repository. It can be accessed at the figshare site 10.6084/m9.figshare.7695662, and we have included reference to this in our supplementary information at the end of the manuscript (P15L3).

[revised manuscript text omitted]